# Influence Analysis of Geometric Error and Compensation Method for Four-Axis Machining Tools with Two Rotary Axes

Guojuan Zhao [1], Shengcheng Jiang [2], Kai Dong [2], Quanwang Xu [1], Ziling Zhang [1] and Lei Lu [2,3,4,*]

1   Logistics Engineering College, Shanghai Maritime University, Shanghai 201306, China; gjzhao070906@126.com (G.Z.); 202030210016@stu.shmtu.edu.cn (Q.X.); zhangziling1119@126.com (Z.Z.)
2   Boneng Transmission (Suzhou) Co., Ltd., Suzhou 215021, China; jesec2006@163.com (S.J.); dkmech@163.com (K.D.)
3   Jiangsu Provincial Key Laboratory of Advanced Robotics, Soochow University, Suzhou 215021, China
4   Collaborative Innovation Center of Suzhou Nano Science and Technology, Soochow University, Suzhou 215021, China
*   Correspondence: jluc2ll@163.com

**Abstract:** Four-axis machine tools with two rotary axes are widely used in the machining of complex parts. However, due to an irregular kinematic relationship and non-linear kinematic function with geometric error, it is difficult to analyze the influence the geometry error of each axis has and to compensate for such a geometry error. In this study, an influence analysis method of geometric error based on the homogeneous coordinate transformation matrix and a compensation method was developed, using the Newton iterative method. Geometric errors are characterized by a homogeneous coordinate transformation matrix in the proposed method, and an error matrix is integrated into the kinematic model of the four-axis machine tool as a means of studying the influence the geometric error of each axis has on the tool path. Based on the kinematic model of the four-axis machine tool considering the geometric error, a comprehensive geometric error compensation calculation model based on the Newton iteration was then constructed for calculating the tool path as a means of compensating for the geometric error. Ultimately, the four-axis machine tool with a curve tool path for an off-axis optical lens was chosen for verification of the proposed method. The results showed that the proposed method can significantly improve the machining accuracy.

**Keywords:** geometric error; error compensation; homogeneous coordinate transformation matrix; Newton iteration; four-axis machining tools




## 1. Introduction

Multi-axis CNC machine tools have high efficiency and exhibit excellent performance. They are widely used in manufacturing, particularly for complex surface machining tasks, and have become a crucial part of modern manufacturing equipment [1,2]. Machining accuracy is essential for the evaluation of machine tool performance. It is affected by geometry, heat, motion, stiffness, vibration, and several other factors. Wu et al. [3] proposed a robust design method for optimizing the static accuracy of a vertical machining center to make the machining accuracy meet design requirements. Niu et al. [4] provided a new analysis method for evaluating machining accuracy reliability based on the nonlinear correlation between errors. Li et al. [5] overviewed the thermal error modeling methods that had been researched and applied in the past ten years. Geometric error is a factor that has a significant impact on machining accuracy and accounts for approximately 40% of all errors. To improve the accuracy of error recognition, Wei et al. [6] provided an overview of the current research algorithms. Lin et al. [7] provided a geometric error modeling method for five-axis CNC machine tools based on the differential transformation method. Geng et al. [8] summarized state-of-the-art research in the calibration of geometric errors of ultra-precision machine tools (UPMTs). Compared to the mature method for traditional

precision machine tools, the increasing use of UPMTs has shown different characteristics in error modeling, measurement, and compensation. In an attempt to improve the machining accuracy of machine tools, several scholars have conducted extensive in-depth research and proposed a variety of effective methods. These methods can be divided into two categories: error prevention methods and error compensation methods. Error prevention methods involve the improvement of the machining and assembly accuracy of machine tool parts [9]. They have great limitations and are quite expensive in economic terms. Error compensation methods offset the original error with an artificially created error as a means of improving machining accuracy. Error compensation mainly includes error detection, error source analysis, accurate error model establishment, error compensation, and effect evaluation. The earliest error compensation technology was used for calculating the relative offset between the actual detection point and the ideal point on the coordinate measuring machine by using rigid body kinematics and small angle assumption [10].

Geometric error modeling and the analysis of machine tools have attracted significant attention and development. Geometric error modeling methods have been developed from a variety of perspectives and include the triangular geometry method [11], rigid body kinematics [12], Denavit–Hardenberg (D–H) method [13], error matrix method [14], and quadratic analysis method [15]. Ding et al. [15] proposed an inverse kinematic method that could compensate for geometric errors. A cutting experiment on a compensated five-axis machine tool was conducted to provide validation of the feasibility of the method. Ding et al. [16] suggested a computational method for geometric error definition and modeling for the reconfigurable machine tool. A coding method for machine components was also presented for automatic error definition and modeling. Maeng et al. [17] used an on-machine measurement method for identifying the geometric errors of rotary axis and tool setting, with previous approaches not having considered errors induced in tool setting. A simulation was conducted as a means of checking the sensitivity of the method, and the model was validated in the experiments. Niu et al. [18] designed a novel Global Sensitivity Analysis (GSA) method and established a spatial error model for analyzing the local influence geometric error had on machining accuracy. Improvement measures were ultimately proposed for verifying the correctness of the method, taking a machining center as an example. Manikandan et al. [19] proposed a mathematical model for the estimation of geometric errors during the turning of a thin-walled hollow cylinder. They believed the study would provide precautionary measures for the more effective and reliable control of dimensional and geometric errors. Fan et al. [20] modeled calculation analysis for a certain type of CNC internal cylindrical compound grinding machine in order to clarify the degree of influence each error parameter had on grinding accuracy. Li et al. [21] developed a precise "ball-column" device for measuring the geometric error of the two rotary axes of the five-axis machine tool. The results showed that the accuracy of the developed error measurement device reached 91.8%, and the measuring time was 30–40 min. Zhong et al. [22] developed a volumetric error model that was based on screw theory, and the identification method for the squareness errors was designed based on the theory considering a three-axis horizontal machine tool. Song et al. [23] presented a high-efficient calculation method for sensitive position-dependent geometric error identification for the five-axis machine tool. In this paper, a series of points in the machining area were selected to compare the machining errors before and after error compensation, and the results show that this method was accurate.

Prior to the implementation of error compensation, calculating the error compensation value is of great importance. Commonly used methods are the iterative method and differential method. The iterative method has been widely studied due to its high accuracy and adaptability. Tang et al. [24] compensated the geometric errors for the accurate worm grinding of spur face gears in order to solve the problem of the cutter rotation angle error not being compensated for in previous compensation methods. They validated the proposed method by using theoretical calculation and practical machining. Ding et al. [25] focused on the identification of the geometric errors for three-axis machine tools, as this is

beneficial for the efficiency and precision of the remanufacturing process. Liang et al. [26] provided a novel compensation method that combined both position and posture errors of the tool center point of an RLLLR five-axis machine tool. As a side note, the RLLLR five-axis machine tool is constructed by two rotatory axes and three linear axes; the kinematic chain from workpiece coordinate system to the tool coordinate system is the rotatory axis, the linear axis, the linear axis, the linear axis, and the rotatory axis successively. Moreover geometric accuracy improvements were enabled by the proposed method. Zha et al. [27] provided a geometric error measurement and compensation method by using a laser tracer for the machining tool, and a complex workpiece was machined that could verify compensating accuracy. Nagayama et al. [28] proposed a deterministic process flow, and geometric error was compensated for before machining, using the proposed method. The results found the form error following compensation to be improved. Lu et al. [29] found an effective way to identify the dominant errors and perform targeted compensation. The models were applied in a real-time compensation system, and the results show that the proposed method could help figure out the most dominant errors and reduce ~90% of the total error. Zhang et al. [30] designed a multi-sensor system consisting of a touch-trigger probe and a laser displacement sensor for enhancing measurement accuracy and efficiency. Fujimori et al. [31] discussed numerical error compensation techniques for geometric error on high-precision machine tools. The error compensation experiments were performed on a linear machine tool. Lu et al. [32] suggested a software-based method for compensating volumetric errors and modifying CNC part programs through the application of previously obtained volumetric error tables to modify the commands. Experiments were conducted, and the rate of reduction was found to be 77.99% for a tested circular contour and 87.59% for a tested spiral contour. Luo et al. [33] provided an intelligent model for a vertical high-speed CNC machine tool spindle by optimizing the temperature measuring points and using artificial neural network technology.

The independently developed four-axis machine tool was taken as the research object for this paper. In the traditional five-axis machine tool compensating process, the position error of tool tip is compensated by the linear axes of the machine tool, and the orientation error of the cutting tool is compensated for by the rotating axes of the machine tool. However, for four-axis machine tools with two rotatory axes, it is difficult to analyze the influence the geometry error of each axis has and to compensate the geometry error, considering its irregular kinematic relationship compared with the traditional five-axis machine tools.

Due to the fact that kinematic model of machine tools with geometric error is a complex highly nonlinear equation, the Newton iterative method is used for solving nonlinear equations. As the irregular kinematic relationship, the Jacobian matrix is a non-spare matrix that increases the difficulty of the iteration process. To solve this issue smoothly, the pseudo-inverse matrix of the Jacobian matrix is used to find the near solution. At the same time, the theoretical coordinates of each axis that are obtained from the inverse solution of the theoretical kinematic model are taken as the initial value, and the Jacobian matrix of the theoretical kinematic model of the machine tool replaces the Jacobian matrix based on geometric error, thereby reducing calculation difficulty. The four-axis machining platform with two linear motors and direct drive turntables is used for verifying the comprehensive error analysis and compensation method, significantly improving machining accuracy. The rest of this paper is organized as follows: In Section 2, the theoretical kinematic relationship for four-axis machine tools is provided. The geometric error is modeled in Section 3. In Section 4, the actual kinematic relationship is established with the geometric errors. The geometric error is compensated for based on the Newton iteration in Section 5. The experiment and simulation of error compensation are presented in Section 6, and finally the paper is concluded in Section 7.

The flowchart is also provided in Figure 1. From the flowchart, we can see that the laser interferometer is used firstly for detecting and analyzing the position of geometric errors of the axes of the four-axis machining platform through many experiments. Secondly, the

multi-body system theory is used for integrating the error matrix into the kinematic model of the machine tool as a means of studying the influence the geometric error of each axis has on the tool path. The nonlinear coupling characteristics of the transformation matrix from the tool coordinate system to workpiece coordinate system can then be analyzed. Finally, a comprehensive error kinematic model of the four-axis machining platform considering geometric error is established.

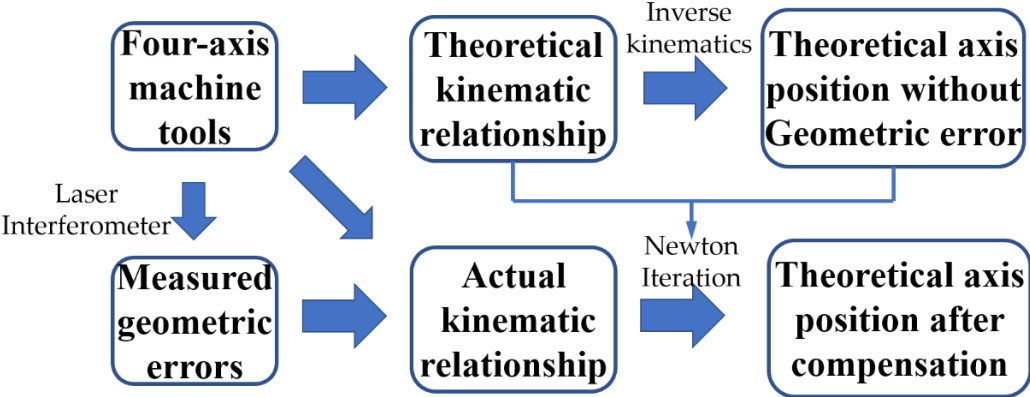

**Figure 1.** The flowchart of the proposed method.

## 2. Theoretical Kinematic Relationship for Four-Axis Machine Tools

The structural model of the four-axis machine tool in this paper can be seen in Figure 2a. It adopts an RTTR configuration with two translation axes (***X***-axis and ***Z***-axis) and two rotation axes (***B***-axis and ***C***-axis). Without the geometric error being considered, two coordinate systems are added to each moving axis; that is, a base coordinate system is fixedly connected with the base of the moving axis, and a moving coordinate system is fixedly connected with the moving part of the moving axis. The direction of the coordinate axis is consistent with the moving direction of each machine tool axis; that is, the ***X*** axis of all coordinate systems is consistent with the moving direction of the ***X*** linear moving axis. The tool coordinate system is attached to the tool tip of the cutting tool, and the workpiece coordinate system is made to coincide with the moving parts of the ***C***-axis coordinate system.

The theoretical kinematic coordinate system model of the four-axis machine tool structure can be seen in Figure 2b. The sequence of the kinematic chain is from the tool coordinate system, $T_t$, to ideal moving coordinate system, $T_{bi}$, of the ***B***-axis, and then to the basic coordinate system, $T_{bo}$, of the ***B***-axis, and so on, until we reach the workpiece coordinate, $T_w$. The complete kinematic chain without the consideration of geometric error is as follows:

$$T_t \Rightarrow T_{bi} \Rightarrow T_{bo} \Rightarrow T_{zi} \Rightarrow T_{zo} \Rightarrow T_{xo} \Rightarrow T_{xi} \Rightarrow T_{co} \Rightarrow T_{ci} \Rightarrow T_w \tag{1}$$

where $T_w$ is the workpiece coordinate system; $T_t$ is the tool coordinate system; $T_{bo}$, $T_{zo}$, $T_{xo}$, and $T_{co}$ form the basic coordinate system of the ***B***, ***Z***, ***X***, and ***C*** axes; and $T_{bi}$, $T_{zi}$, $T_{xi}$, and $T_{ci}$ are the moving coordinate system of the ***B***, ***Z***, ***X***, and ***C*** axes.

According to the kinematic structure in Figure 2b and actual measurements taken in the laboratory, the distance from the tool-tip point to the origin of $T_{bi}$ along the ***Y***-axis is $d_1 = 100$ mm; the distance from the tool-tip point to the origin of $T_{bi}$ along the ***Z***-axis is $L_1 = 250$ mm; the distance from the origin of $T_{bo}$ to the origin of $T_{zi}$ along the ***Y***-axis is $d_2 = 100$ mm; the distance from the origin of $T_{zo}$ to the origin of $T_{xo}$ along the ***Z***-axis is $L_2 = 360$ mm; the distance from the origin of $T_{zo}$ to the origin of $T_{xo}$ along the ***Z***-axis is $d_3 = 50$ mm; the distance from the origin of $T_{xi}$ to the origin of $T_{co}$ along the ***Y***-axis is $d_4 = 150$ mm; and the distance from the origin of $T_{xi}$ to the origin of $T_{co}$ along the ***Z***-axis is $L_3 = 150$ mm.

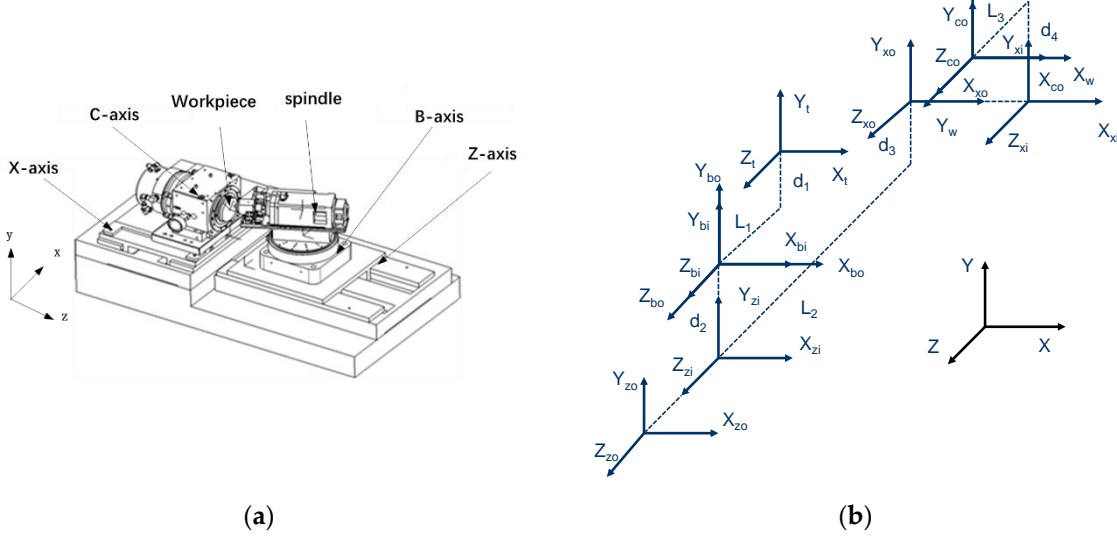

**Figure 2.** Structural diagram of the four-axis machine tool. (**a**) The structure model of the four-axis machine tool. (**b**) The coordinate system of kinematic structure of the four-axis machine tool.

Based on the constructed coordinate systems of each axis and the constants between coordinate systems and motion variables $[X, Z, B, C]$ of each axis, the homogeneous coordinate transformation matrix, $^{w}T_t$, between the tool coordinate system, $T_t$, and workpiece coordinate system, $T_w$, without a geometric error is obtained as follows:

$$^{w}T_t(X,Z,B,C) = {}^{w}T_{ci}{}^{ci}T_{co}(C){}^{co}T_{xi}{}^{xi}T_{xo}(X){}^{xo}T_{zo}{}^{zo}T_{zi}(Z){}^{zi}T_{bo}{}^{bo}T_{bi}(B){}^{bi}T_t \qquad (2)$$

where we have the following:

$$^{w}T_{ci} = \begin{bmatrix} 1 & 0 & 0 & 0 \\ 0 & 1 & 0 & 0 \\ 0 & 0 & 1 & 0 \\ 0 & 0 & 0 & 1 \end{bmatrix}$$

$$^{ci}T_{co}(C) = \begin{bmatrix} \cos(C) & \sin(C) & 0 & 0 \\ -\sin(C) & \cos(C) & 0 & 0 \\ 0 & 0 & 1 & 0 \\ 0 & 0 & 0 & 1 \end{bmatrix}$$

$$^{co}T_{xi} = \begin{bmatrix} 1 & 0 & 0 & 0 \\ 0 & 1 & 0 & -d_4 \\ 0 & 0 & 1 & -L_3 \\ 0 & 0 & 0 & 1 \end{bmatrix}$$

$$^{xi}T_{xo}(X) = \begin{bmatrix} 1 & 0 & 0 & -X \\ 0 & 1 & 0 & 0 \\ 0 & 0 & 1 & 0 \\ 0 & 0 & 0 & 1 \end{bmatrix}$$

$$^{xo}T_{zo} = \begin{bmatrix} 1 & 0 & 0 & 0 \\ 0 & 1 & 0 & -d_3 \\ 0 & 0 & 1 & L_2 \\ 0 & 0 & 0 & 1 \end{bmatrix}$$

$$
{}^{zo}T_{zi}(Z) = \begin{bmatrix} 1 & 0 & 0 & 0 \\ 0 & 1 & 0 & 0 \\ 0 & 0 & 1 & Z \\ 0 & 0 & 0 & 1 \end{bmatrix}
$$

$$
{}^{zi}T_{bo} = \begin{bmatrix} 1 & 0 & 0 & 0 \\ 0 & 1 & 0 & d_2 \\ 0 & 0 & 1 & 0 \\ 0 & 0 & 0 & 1 \end{bmatrix}
$$

$$
{}^{bo}T_{bi}(B) = \begin{bmatrix} \cos(B) & 0 & \sin(B) & 0 \\ 0 & 1 & 0 & 0 \\ -\sin(B) & 0 & \cos(B) & 0 \\ 0 & 0 & 0 & 1 \end{bmatrix}
$$

$$
{}^{bi}T_t = \begin{bmatrix} 1 & 0 & 0 & 0 \\ 0 & 1 & 0 & d_1 \\ 0 & 0 & 1 & -L_1 \\ 0 & 0 & 0 & 1 \end{bmatrix}
$$

In the tool coordinate system, $T_t$, the coordinate position vector of the tool-tip point, $P_t$, and the tool axis direction, $O_t$, are constants, and they are generally expressed in the following way in homogeneous coordinates:

$$
P_t = [0, 0, 0, 1]^T,\ O_t = [0, 0, -1, 0]^T \tag{3}
$$

According to the machining process and the surface that is to be machined, the tool pose vector $\{P_w, O_w\}$ in the workpiece coordinate system $T_w$ is as follows:

$$
P_w = [x, y, z, 1]^T,\ O_w = [i, j, k, 0]^T \tag{4}
$$

Therefore, the axis variables $[X, Z, B, C]$ of the four-axis machine tool should be driven to ensure that the tool position and vector satisfy designed tool pose vector $\{P_w, O_w\}$ in $T_w$:

$$
\begin{cases} P_w = {}^{w}T_t(X,Z,B,C)\cdot P_t \\ O_w = {}^{w}T_t(X,Z,B,C)\cdot O_t \end{cases} \tag{5}
$$

Equation (5) is the theoretical kinematic equation for the four-axis machine tools without consideration of the geometric error. If axis position $(X, Z, B, C)$ is given, the corresponding tool pose vector $\{P_w, O_w\}$ in $T_w$ can be calculated. According to the kinematic relation, the equation has four axis variables, and the three dimensions of tool-path-position-following requirements should be satisfied. Therefore, the tool axis vector only has one degree of freedom, and the tool axis should be limited on a determined cone. Equation (5) is then full rank and has a finite number of solutions.

### 3. Modeling the Geometric Error

Theoretically, each kinematic pair of machining axis only has one degree of freedom in the given direction of motion. However, motion is different in the actual state, and there are motion geometric errors of six degrees of freedom. The error of any kinematic pair is assumed to not affect other pairs. There are six elements to the motion error of any motion pair and the movement and rotation relative to the *X*, *Y*, and *Z* axes, which are the geometric errors of machine tools, mainly include positioning error, straightness error, rolling angle error, and yaw angle error.

Figure 3 shows the geometric error element in relation to the position. When the moving part moves along the *X*-axis direction, there are three translational motion error components: linear displacement error, $\delta_x(x)$; straightness error in *Y* direction, $\delta_y(x)$; and straightness error in *Z* direction, $\delta_z(x)$. There are also three angular motion error compo-

nents: rolling angle error, $\varepsilon_x(x)$; pitch angle error, $\varepsilon_y(x)$; and yaw angle error, $\varepsilon_z(x)$, where $x$, $y$, and $z$ represent error motion direction. These errors are only related to the position of the motion axis and do not relate to the position of other axes. Figure 3b demonstrates that the rotating pair rotates $\theta$ degree around the Z-axis of the rotation axis in coordinate system *O-XYZ*. There are six error motion components: radial error along the X-axis, $\delta_x(\theta)$; radial error along the Y-axis, $\delta_y(\theta)$; axial error along the Z-axis, $\delta_z(\theta)$; the errors of two inclination angles, $\varepsilon_x(\theta)$ and $\varepsilon_y(\theta)$; and positioning error, $\varepsilon_z(\theta)$.

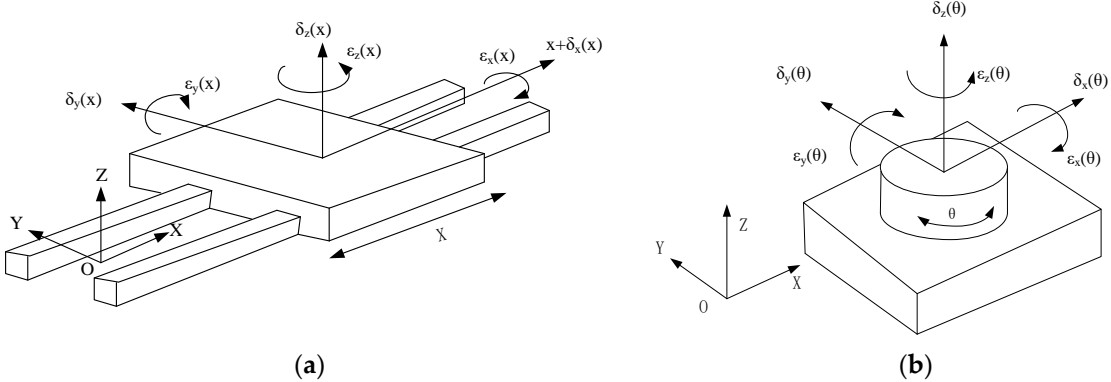

**Figure 3.** Geometric error element in relation to the position. (**a**) Geometric error element of movement pair. (**b**) Geometric error element of rotation pair.

The geometric error of the machine tool is the difference between the actual position and theoretical position of the moving parts of each moving axis. With the Renishaw laser interferometer measurement method [34], one lens group is installed on the mobile platform, and the other is installed on the basic bed. The machine tool controller sends the position command to the moving axis and the actual position of the moving part of the axis is measured by the mirror group that is installed on the moving part of the axis. The geometric error of the moving axis of the machine tool is the difference between actual position and commanded position.

For analyzing and compensating for the geometric error, combining the measured geometric error value with the kinematic equation of the four-axis machine tool is necessary. As the geometric error is the difference between the actual position of moving parts on the machine tool and the command position (theoretical position), the geometric error can be characterized by using the homogeneous coordinate transformation matrix. As can be seen in Figure 4, taking the X-axis as an example, a command coordinate system, $T_{xr}$, representing the actual position of the moving parts of each axis is added to each moving axis, with the exception of the base coordinate system, $T_{xo}$, and moving coordinate system, $T_{xi}$, which represent the theoretical position. The static mirror group of the Renishaw laser interferometer is generally installed on the reference plane of the moving axis, whereas the moving mirror group of the Renishaw laser interferometer is installed on the moving part of the moving axis. Data obtained through the geometric error measurement by Renishaw laser interferometer is then the relative position relationship between theoretical position, $T_{xi}$, of the motion axis and actual position coordinate system, $T_{xr}$, of the motion axis.

Therefore, the transformation matrix, $^{xr}T_{xi}$, between the theoretical position, $T_{xi}$, of the motion axis and actual position coordinate system, $T_{xr}$, of the motion axis can be obtained by the measured geometric error, $E_x = \left[\delta_{xx}, \delta_{yx}, \delta_{zx}, \varepsilon_{xx}, \varepsilon_{yx}, \varepsilon_{zx}\right]$. Firstly, as the geometric error is small, and ignoring the high-order parts, the transformation matrix, $^{xr}T_{xi}$, is constructed as follows:

$$E_x = \left[\delta_{xx}, \delta_{yx}, \delta_{zx}, \varepsilon_{xx}, \varepsilon_{yx}, \varepsilon_{zx}\right] \Rightarrow {}^{xr}T_{xi}(X) = \begin{bmatrix} 1 & -\varepsilon_{zx} & \varepsilon_{yx} & \delta_{xx} \\ \varepsilon_{zx} & 1 & -\varepsilon_{xx} & \delta_{yx} \\ -\varepsilon_{yx} & \varepsilon_{xx} & 1 & \delta_{zx} \\ 0 & 0 & 0 & 1 \end{bmatrix} \tag{6}$$

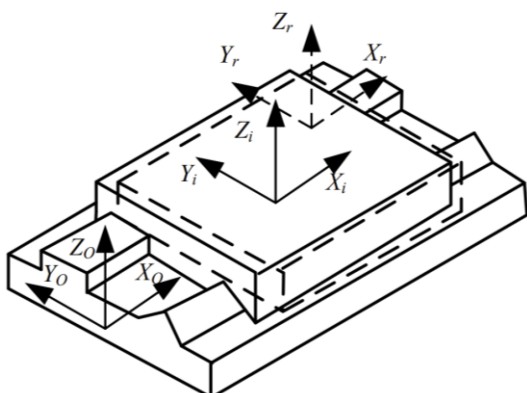

**Figure 4.** Schematic diagram of coordinate system.

## 4. Actual Kinematic Relationship with Geometric Error

Due to the existence of geometric error, the motion of each axis that is solved by using the theoretical kinematic model causes the machine tool to deviate from the designed tool position and posture. Therefore, calculating the influence of geometric error on motion trajectory is necessary. The actual kinematic model of the machine tool that considers geometric error should be constructed. Firstly, the actual complete kinematic chain that considers the geometric error is as follows:

$$T_t \Rightarrow T_{br} \Rightarrow T_{bi} \Rightarrow T_{bo} \Rightarrow T_{zr} \Rightarrow T_{zi} \Rightarrow T_{zo} \Rightarrow T_{xo} \Rightarrow T_{xi} \Rightarrow T_{xr} \Rightarrow T_{co}$$
$$\Rightarrow T_{ci} \Rightarrow T_{cr} \Rightarrow T_w. \tag{7}$$

Based on the constructed coordinate systems of each axis, constants between coordinate systems and motion variables $[X, Z, B, C]$ of each axis, the homogeneous coordinate transformation matrix ${}^wT_t(X, Z, B, C)_e$ between the tool coordinate system, $T_t$, and workpiece coordinate system, $T_w$, without geometric error is obtained as follows:

$${}^wT_t(X, Z, B, C)_e = {}^wT_{cr}{}^{cr}T_{ci}(C){}^{ci}T_{co}(C){}^{co}T_{xr}{}^{xr}T_{xi}(X){}^{xi}T_{xo}(X){}^{xo}T_{zo}{}^{zo}T_{zi}(Z){}^{zi}T_{zr}(Z){}^{zr}T_{bo}{}^{bo}T_{bi}(B){}^{bi}T_{br}(B){}^{br}T_t \tag{8}$$

where

$${}^wT_{cr} = \begin{bmatrix} 1 & 0 & 0 & 0 \\ 0 & 1 & 0 & 0 \\ 0 & 0 & 1 & 0 \\ 0 & 0 & 0 & 1 \end{bmatrix}$$

$${}^{cr}T_{ci}(C) = \begin{bmatrix} 1 & -\varepsilon_{zc} & \varepsilon_{yc} & \delta_{xc} \\ \varepsilon_{zc} & 1 & -\varepsilon_{xc} & \delta_{yc} \\ -\varepsilon_{yc} & \varepsilon_{xc} & 1 & \delta_{zc} \\ 0 & 0 & 0 & 1 \end{bmatrix}$$

$${}^{ci}T_{co}(C) = \begin{bmatrix} \cos(C) & \sin(C) & 0 & 0 \\ -\sin(C) & \cos(C) & 0 & 0 \\ 0 & 0 & 1 & 0 \\ 0 & 0 & 0 & 1 \end{bmatrix}$$

$${}^{co}T_{xr} = \begin{bmatrix} 1 & 0 & 0 & 0 \\ 0 & 1 & 0 & -d_4 \\ 0 & 0 & 1 & -L_3 \\ 0 & 0 & 0 & 1 \end{bmatrix}$$

$$
{}^{xr}T_{xi}(X) = \begin{bmatrix} 1 & -\varepsilon_{zx} & \varepsilon_{yx} & \delta_{xx} \\ \varepsilon_{zx} & 1 & -\varepsilon_{xx} & \delta_{yx} \\ -\varepsilon_{yx} & \varepsilon_{xx} & 1 & \delta_{zx} \\ 0 & 0 & 0 & 1 \end{bmatrix}
$$

$$
{}^{xi}T_{xo}(X) = \begin{bmatrix} 1 & 0 & 0 & -X \\ 0 & 1 & 0 & 0 \\ 0 & 0 & 1 & 0 \\ 0 & 0 & 0 & 1 \end{bmatrix}
$$

$$
{}^{xo}T_{zo} = \begin{bmatrix} 1 & 0 & 0 & 0 \\ 0 & 1 & 0 & -d_3 \\ 0 & 0 & 1 & L_2 \\ 0 & 0 & 0 & 1 \end{bmatrix}
$$

$$
{}^{zo}T_{zi}(Z) = \begin{bmatrix} 1 & 0 & 0 & 0 \\ 0 & 1 & 0 & 0 \\ 0 & 0 & 1 & Z \\ 0 & 0 & 0 & 1 \end{bmatrix}
$$

$$
{}^{zi}T_{zr}(Z) = \begin{bmatrix} 1 & -\varepsilon_{zz} & \varepsilon_{yz} & \delta_{xz} \\ \varepsilon_{zz} & 1 & -\varepsilon_{xz} & \delta_{yz} \\ -\varepsilon_{yz} & \varepsilon_{xz} & 1 & \delta_{zz} \\ 0 & 0 & 0 & 1 \end{bmatrix}
$$

$$
{}^{zr}T_{bo} = \begin{bmatrix} 1 & 0 & 0 & 0 \\ 0 & 1 & 0 & d_2 \\ 0 & 0 & 1 & 0 \\ 0 & 0 & 0 & 1 \end{bmatrix}
$$

$$
{}^{bo}T_{bi}(B) = \begin{bmatrix} \cos(B) & 0 & \sin(B) & 0 \\ 0 & 1 & 0 & 0 \\ -\sin(B) & 0 & \cos(B) & 0 \\ 0 & 0 & 0 & 1 \end{bmatrix}
$$

$$
{}^{bi}T_{br}(B) = \begin{bmatrix} 1 & -\varepsilon_{zb} & \varepsilon_{yb} & \delta_{xb} \\ \varepsilon_{zb} & 1 & -\varepsilon_{xb} & \delta_{yb} \\ -\varepsilon_{yb} & \varepsilon_{xb} & 1 & \delta_{zb} \\ 0 & 0 & 0 & 1 \end{bmatrix}
$$

$$
{}^{br}T_{t} = \begin{bmatrix} 1 & 0 & 0 & 0 \\ 0 & 1 & 0 & d_1 \\ 0 & 0 & 1 & -L_1 \\ 0 & 0 & 0 & 1 \end{bmatrix}
$$

The axis variables [$X, Z, B, C$] of the four-axis machine tool are calculated as follows when the geometric error is considered:

$$
\begin{cases} P_w = {}^{w}T_t(X, Z, B, C)_e \cdot P_t \\ O_w = {}^{w}T_t(X, Z, B, C)_e \cdot O_t \end{cases} \tag{9}
$$

The influence of the geometric error can then be analyzed by using Equation (9); as axis position $(X, Z, B, C)$ is given, the corresponding tool-pose vector $\{P_w, O_w\}$ in $T_w$ can also be calculated when the geometric error is considered. At the same time, if the geometric error is obtained and the $\{P_w, O_w\}$ in the workpiece space are given, the actual position [$X, Z, B, C$] of the machine tool can be calculated, and the geometric errors can be compensated.

## 5. Compensate Geometric Error Based on the Newton Iteration

As the geometric error is position-dependent, the actual position [*X, Z, B, C*] that is calculated from the kinematic relation when geometric errors are considered is difficult to obtain. A numerical calculation method based on the Newton iteration method was designed in this study to obtain the accurate position [*X, Z, B, C*] from the kinematic relation with the geometric errors.

### 5.1. Newton Iteration Method for Solving Nonlinear Equations

Newton iteration is a method that can be used for solving nonlinear equations by linearizing the function $f(x)$ locally with the derivative value as the slope. The slope value is continuously modified through continuous iteration until the solution of $f(x) = 0$ is found. For nonlinear equation $f(x) = 0$, the solution formula [35] is as follows:

$$x_{k+1} = x_k - \frac{f(x_k)}{f'(x_k)}, \ k = 0, \ 1 \ldots \tag{10}$$

For a group of nonlinear equations, it is as follows:

$$\begin{cases} f_1(x_1, x_2, \cdots x_n) = 0 \\ f_2(x_1, x_2, \cdots x_n) = 0 \\ \qquad\qquad \vdots \\ f_n(x_1, x_2, \cdots x_n) = 0 \end{cases} \tag{11}$$

where $f_1 \ f_2 \cdots \ f_n$ is the multivariate function of $x_1, x_2, \cdots x_n$. In matrix form, Equation (11) is represented as $\boldsymbol{F}(\boldsymbol{x}) = 0$, where $\boldsymbol{x} = [x_1, x_2, \cdots x_n]$, $\boldsymbol{x} \in \boldsymbol{R^n}$, and $\boldsymbol{F} = [f_1 \ f_2 \ \cdots \ f_n]$.

The Newton iterative method for solving nonlinear equation groups is extended as follows:

$$\boldsymbol{x}^{(k+1)} = \boldsymbol{x}^{(k)} - \boldsymbol{F}'\left(\boldsymbol{x}^{(k)}\right)^{-1} \boldsymbol{F}\left(\boldsymbol{x}^{(k)}\right), \ k = 0, 1 \ldots \tag{12}$$

where $\boldsymbol{F}'(\boldsymbol{x})$ is the Jacobian matrix for $\boldsymbol{F}(\boldsymbol{x})$:

$$\boldsymbol{F}'(\boldsymbol{x}) = \begin{bmatrix} \frac{\partial f_1}{\partial x_1} & \cdots & \frac{\partial f_1}{\partial x_n} \\ \vdots & \ddots & \vdots \\ \frac{\partial f_n}{\partial x_1} & \cdots & \frac{\partial f_n}{\partial x_n} \end{bmatrix} \tag{13}$$

When number *m* of equations *f* is greater than number *n* of unknowns $x_n$, i.e., $m > n$, Equation (13) of nonlinear equations has no solution, but the generalized inverse matrix for $\boldsymbol{F}'(\boldsymbol{x})$ is constructed for finding the solution in the sense of least squares. (To be noted that, the solution found by this method is the near solution, as the $\boldsymbol{F}(\boldsymbol{x})_{m \times n}$ has no solution when $m > n$). The generalized inverse matrix $\boldsymbol{F}'(\boldsymbol{x})^+$ is as follows:

$$\boldsymbol{F}'(\boldsymbol{x})^+ = \left(\boldsymbol{F}'(\boldsymbol{x})^T \boldsymbol{F}'(\boldsymbol{x})\right)^{-1} \boldsymbol{F}'(\boldsymbol{x})^T \tag{14}$$

### 5.2. Error Compensation Method Based on the Newton Iteration

Considering the nonlinearity and complexity of the kinematic relation with the geometric error, as shown in Equation (9), the kinematic equation can be solved by a numerical calculation method that is based on the Newton iteration. Equation (9) is then changed to the following:

$$\boldsymbol{F_e}(\boldsymbol{x}) = \begin{cases} P_w - {}^w T_t(X, Z, B, C)_e \cdot P_t = 0 \\ O_w - {}^w T_t(X, Z, B, C)_e \cdot O_t = 0 \end{cases} \tag{15}$$

According to the Newton iterative method for solving a group of nonlinear equations, the iterative formula is as follows:

$$x^{(k+1)} = x^{(k)} - F_e'\left(x^{(k)}\right)^{-1} F\left(x^{(k)}\right), \ k = 0, 1 \tag{16}$$

where $F_e'(x)$ is the Jacobian matrix for $F_e(x)$. Although the tool axis vector is designed to ensure that Equation (16) has a finite solution, Equation (16) is non-rank. Jacobian matrix $F_e'(x)$ can then be replaced by $F_e'(x)^+$ according to Equation (14). Jacobian matrix $F_e'(x)$ is the $6 \times 4$ order matrix, as the $P_w$ and $O_w$ are all three-dimensional vectors.

As the position-dependent geometric error is considered in the kinematic equation as shown in Equation (15), calculating $F_e'(x)^+$ is difficult because the geometric error has a non-analytical expression relationship. However, the geometric error is small and can be ignored compared to the machine tool motion in local differentiation. Therefore, in Equation (16), $F_e'(x)^+$ is replaced by the theoretical kinematic relationship $F'(x)^+$ without consideration of geometric error:

$$F_e(x) = \begin{cases} P_w - {}^wT_t(X, Z, B, C)_e \cdot P_t = 0 \\ O_w - {}^wT_t(X, Z, B, C)_e \cdot O_t = 0 \end{cases} \tag{17}$$

$$F'(x) = -\begin{cases} \frac{\partial\, {}^wT_t(X,Z,B,C) \cdot P_t}{\partial X} \\ \frac{\partial\, {}^wT_t(X,Z,B,C) \cdot O_t}{\partial X} \end{cases}$$

$$F'(x)^+ = \left(F'(x)^T F'(x)\right)^{-1} F'(x)^T$$

At the same time, although the geometric error is considered in Equation (17), the solution of Equation (17) is close to the solution of Equation (5). Equation (5) has the analytical-form solution, and this solution can be set as the initial value for the iterative process of Equation (17) for accelerating the solving process.

## 6. Experiment and Simulation of Error Compensation

The four-axis machine tool that is shown in Figure 5 is used for studying the influence of geometric error and the compensation method based on the Newton iteration calculation method. The machine tool combines with two linear axes and two rotatory axes. The two linear axes are ABL8000 Series linear motor stage with air bearing developed by Aerotech. The *B*-axis is ABRS Series air-bearing rotary stage also developed by Aerotech company. The *C*-axis is ultraprecision work-holding spindles (SP150 High performance) developed by Precitech company. Figure 5a shows the machine tool, while Figure 5b shows the machined surface workpiece.

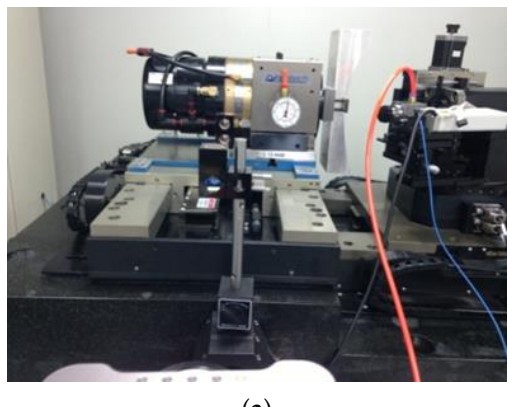 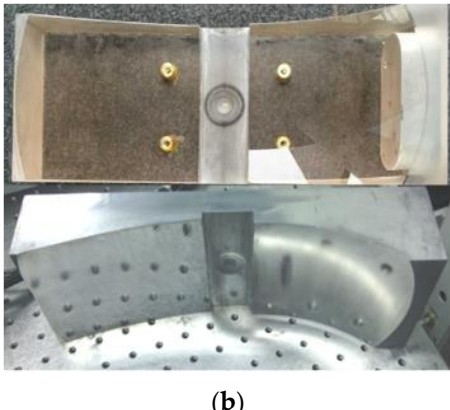

(**a**)                                                           (**b**)

**Figure 5.** Four-axis machine tool and machined surface workpiece. (**a**) Four-axis machine tool. (**b**) Machined surface workpiece.

According to the tool path of the machining process and the designed workpiece, one of the tool paths shown in Figure 6 is selected as the tool path curve of experimental verification algorithm theory for verifying the influence geometric error has on tool path. In the figure, the tool-tip point trajectory is represented by the red point, the tool axis vector is represented by the blue line, and there are 241 tool path points.

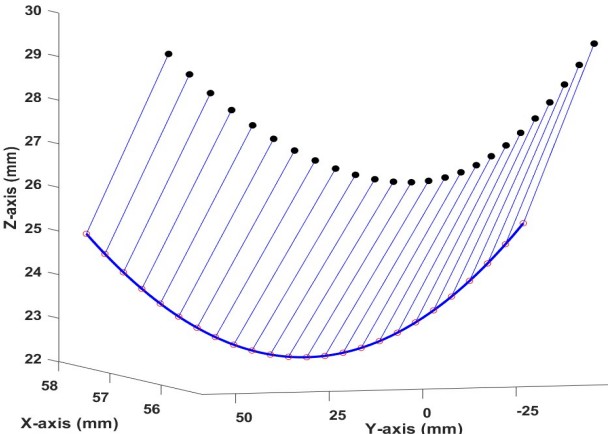

**Figure 6.** Tool-tip point trajectory and axis vector.

According to the theoretical kinematic relationship without the consideration of geometric error, the theoretical axis position can be calculated based on Equation (5). The figures of the axis position series [*X*, *Z*, *B*, *C*] are then presented as in Figure 7:

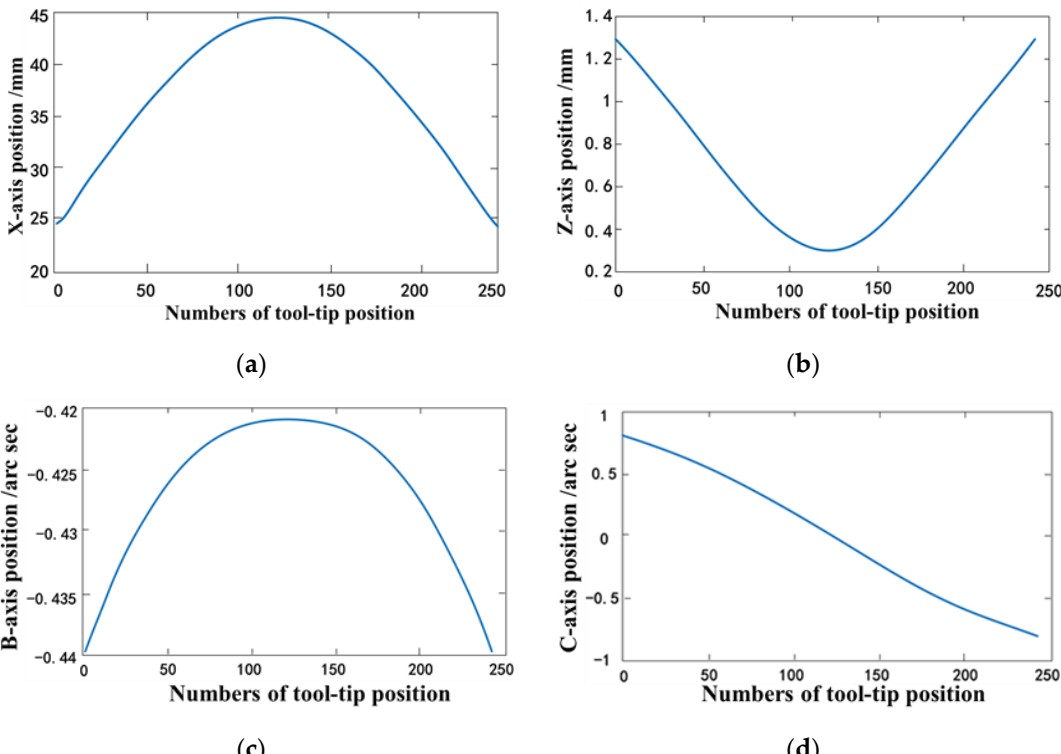

**Figure 7.** Axis position obtained from theoretical kinematic structure: (**a**) *X*-axis position, (**b**) *Z*-axis position, (**c**) *B*-axis position, and (**d**) *C*-axis position.

Due to the existence of geometric error, the position of each axis of the four-axis machine tool obtained by using the theoretical motion model will cause the tool axis vector to deviate from the designed tool path.

### 6.1. Geometric Error Measurements and Influence of the Position of Geometric Error

As Figure 8 shows, the laser interferometer (an XL-80 laser interferometer system of the Renishaw company which has a lens group for measuring linear displacement, speed, angle (pitch and torsion), straightness, flatness, perpendicularity, and parallelism [34]) is used for repeatedly measuring positioning error, straightness error, yaw error, and pitch error for the *X*-axis, *Z*-axis, and *B*-axis based on Standard ISO 230. Due to laboratory conditions, the geometric errors of the *C*-axis are not measured. However, the geometric errors from other axes are compensated by *C*-axis. However, the three motion axes are able to verify the theory that is proposed in this paper.

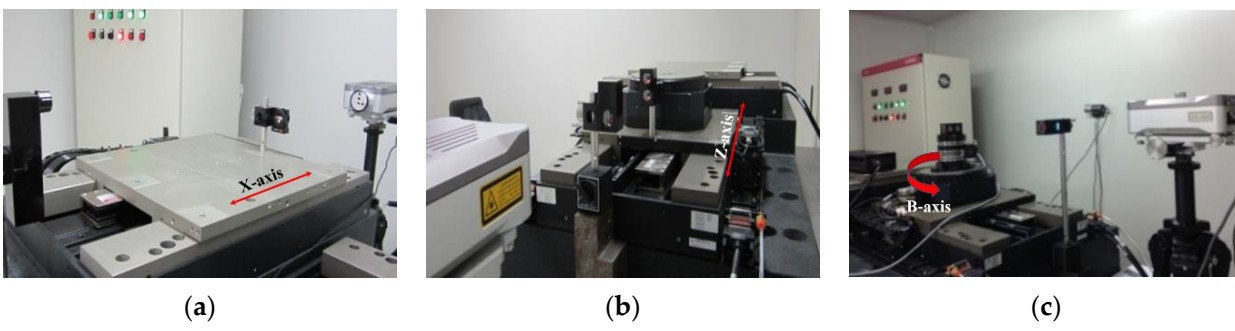

(**a**)　　　　　　　　(**b**)　　　　　　　　(**c**)

**Figure 8.** Measuring process. (**a**) Measuring process for the *Z*-axis. (**b**) Measuring process for the *X*-axis. (**c**) Measuring process for the *B*-axis.

To filter out the random errors, the third-order B-spline is used to fit the geometric error based on the least square. The measured geometric errors and fitted results for the *X*-axis can be seen in Figure 9.

The measured geometric errors and fitted results for the *Z*-axis can be seen in Figure 10.

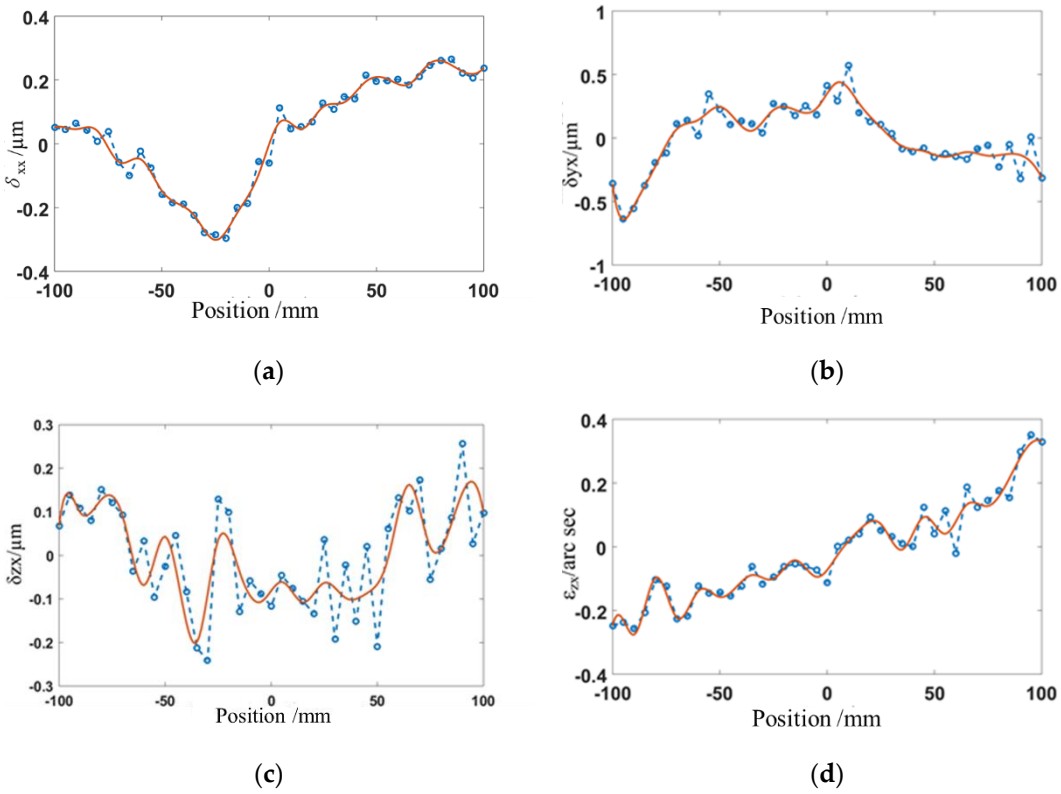

**Figure 9.** *Cont*.

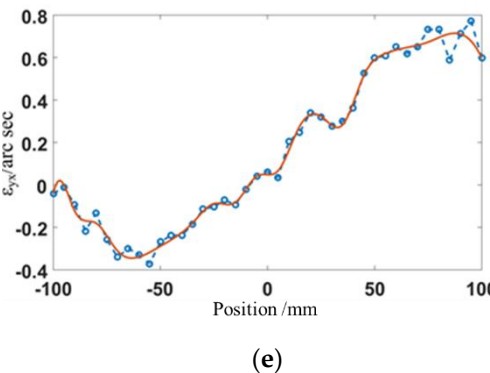

(**e**)

**Figure 9.** Geometric errors for the *X*-axis. (**a**) Linear displacement error $\delta_{xx}(x)$ for the *X*-axis. (**b**) Straightness error in the Y direction $\delta_{yx}(x)$ for the *X*-axis. (**c**) Straightness error in the Z direction $\delta_{zx}(x)$ for the *X*-axis. (**d**) Yaw angle error $\varepsilon_z(x)$ for the *X*-axis. (**e**) Pitch angle error $\varepsilon_y(x)$ for the *X*-axis.

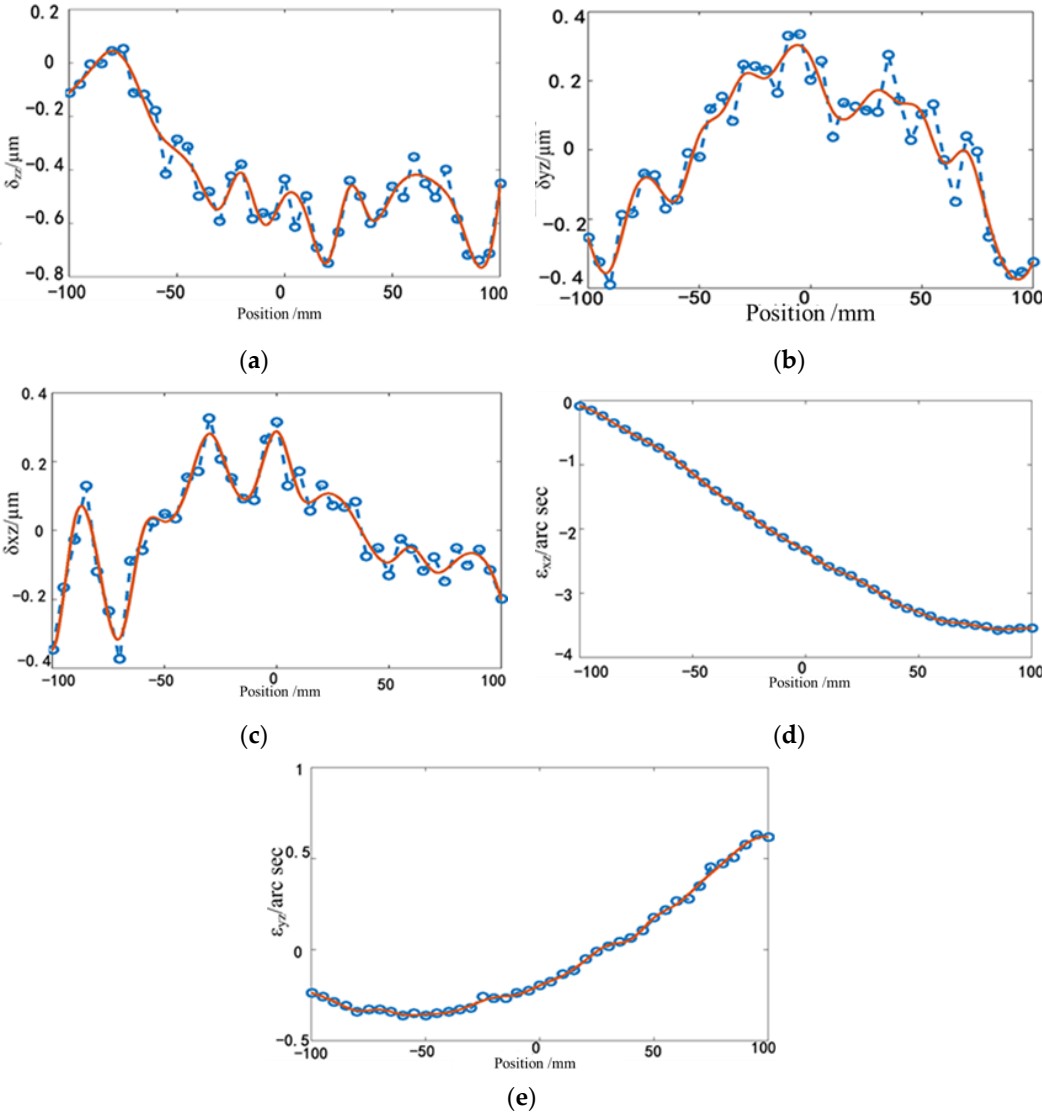

**Figure 10.** Geometric errors for the *Z*-axis. (**a**) Linear displacement error $\delta_{zz}(z)$ for the *Z*-axis; (**b**) Straightness error in the *Y* direction $\delta_{yz}(z)$ for the *Z*-axis. (**c**) Straightness error in the *X* direction $\delta_{xz}(z)$ for the *Z*-axis. (**d**) Yaw angle error $\varepsilon_{xz}(z)$ for the *Z*-axis. (**e**) Pitch angle error $\varepsilon_{yz}(Z)$ for the *Z*-axis.

The rotation displacement error and fitted results for the *B*-axis can be seen in Figure 11.

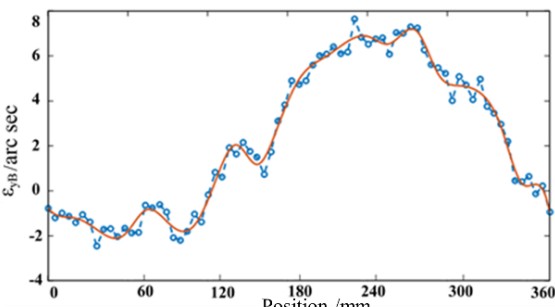

**Figure 11.** Rotation displacement error $\varepsilon_{yB}(B)$ for the *B*-axis.

The fitted geometric error can be used in actual kinematic relation in Equation (15). To reflect the influence of the geometric error, the axis position $P_{th}$ = [$X$, $Z$, $B$, $C$] that is calculated from the theoretical kinematic relation for the tool path in Figure 7 is used in the actual kinematic equation with measured geometric error. When the geometric error of the kinematic relation for the four-axis machine tool is considered, the tool-tip position and orientation deviate from the designed tool position and orientation if $P_{th}$ = [$X$, $Z$, $B$, $C$] is used for controlling the machine tool.

To quantitatively analyze the influence geometric error has on tool position and orientation trajectory, the relative position error, $E$, for tool-tip position and relative angle error, $\varepsilon$, of tool orientation are constructed.

$$E = \|P_e - P_i\| \tag{18}$$

$$\varepsilon = \arccos(O_e - O_i)$$

where $P_i$ and $O_i$ are the design tool tip and tool orientation for the tool path, as shown in Figure 6; $P_e$ and $O_e$ are the tool tip and tool orientation calculated from the theory axis position; and $P_{th}$ = [$X$, $Z$, $B$, $C$] in the actual kinematic relation. The calculated position error, $E$, and relative angle error, $\varepsilon$, for the tool path are shown in Figure 12. From the figures, it can be seen that the tool-tip error is approximately $4 \times 10^{-3}$ mm, and the deviation in tool orientation angle compared to the designed one is approximately $3.5 \times 10^{-4}$ arc sec with the influence of the geometric error. Although these deviations are relatively small for ordinary machine tools, there are large deviations caused by geometric errors for the ultra-precision machine tools that are used in this paper.

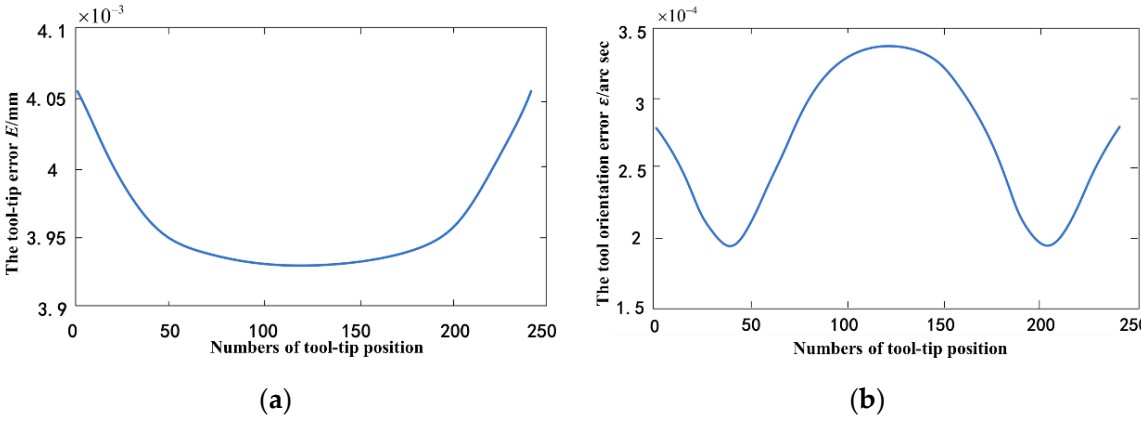

**Figure 12.** Deviation errors of tool-tip position and tool orientation. (**a**) Tool-tip position error. (**b**) Tool axis orientation error.

*6.2. Calculating the Actual Position Considering Geometric Error*

Due to geometric error, the movement of each axis that is obtained from the theoretical kinematic model will cause the tool-tip position and orientation to deviate from the designed tool axis vector. Therefore, solving the position of each moving axis according to the actual kinematic model, as shown in Equation (15), of the four-axis machine tool considering geometric error is necessary.

For the kinematic model without geometric error being considered, an analytical solution exists for the inverse kinematic problem. However, the kinematic model that considers the geometric error is a set of nonlinear equations and no accurate analytical solution exists. Therefore, to obtain the machining path considering geometric error compensation, the Newton iteration is used for solving the actual kinematic equation considering geometric error.

As shown in Equation (15), the actual kinematical equations group

$$F_e(x) = \begin{cases} P_w - {}^wT_t(X,Z,B,C)_e \cdot P_t = 0 \\ O_w - {}^wT_t(X,Z,B,C)_e \cdot O_t = 0 \end{cases}$$

has six equations and four unknown variables.

Therefore, the solution regarding least squares can be obtained, but it is not possible to obtain the exact solution. According to the Newton iterative solution method, the Jacobian matrix of $F(x)$ must be provided. From the above analysis, it can be seen that the Jacobian matrix of $F(x)$ must be replaced by the theoretical kinematic transfer matrix. The Jacobian

matrix $F'(x) = -\begin{cases} \frac{\partial {}^wT_t(X,Z,B,C) \cdot P_t}{\partial X} \\ \frac{\partial {}^wT_t(X,Z,B,C) \cdot O_t}{\partial X} \end{cases}$ of $F(x)$ is a $6 \times 4$ matrix, and this can be calculated by

using the symbolic function of MATLAB *diff* (F,'t'). The pseudo-inverse matrix is then constructed as $F'(x)^+ = \left(F'(x)^T F'(x)\right)^{-1} F'(x)^T$, and $F'(x)^+$ is a $4 \times 6$ matrix. The generalized inverse Newton iterative solution formula of nonlinear equations is then $F(x) = 0$ is

$$x^{(k+1)} = x^{(k)} - F'\left(x^{(k)}\right)^+ F\left(x^{(k)}\right), k = 0, 1 \ldots$$

According to the method, the initial value of the iterative process chooses the $P_{\_th} = [X, Z, B, C]_0$ that is calculated from the theoretical kinematic relation, and the iteration termination condition is $\|x^{(k+1)} - x^{(k)}\| \leq 0$. The actual positions that are calculated by using the kinematic equation that considers the geometric error are then the compensated axis position of the four-axis machine tools for the tool path in the workpiece space, as can be seen in Figure 6. The compensated axis position profiles are displayed in Figure 13, and the trajectory of each axis that is calculated by using the theoretical equation is also provided.

There is a difference between the motion of each axis that is obtained from the kinematic equation considering the geometric error and the motion that is obtained from the theoretical kinematic structure. The difference is the compensation value of each motion axis considering the geometric error, which can be seen in Figure 14. The range of the $X$-axis error compensation values is $3.7 \times 10^{-4} \sim 5.52 \times 10^{-4}$ mm; the range of the $Z$-axis error compensation values is $2.29 \times 10^{-3} \sim 2.361 \times 10^{-3}$ mm; the range of the $B$-axis error compensation values is $-4.79 \times 10^{-6} \sim -3.98 \times 10^{-6}$ arc sec; and the range of the $C$-axis error compensation values is $-4.58 \times 10^{-5} \sim -3.47 \times 10^{-5}$ arc sec.

To verify that the axis position that is obtained by the calculation method based on the Newton iteration can reduce the influence of geometric error, tool-tip point and tool orientation were calculated from the actual kinematic model. Deviations from designed tool-tip position and tool orientation were also obtained. The tool-path deviation for each sampling point can be seen in Figure 15. Compared to Figure 12, the compensated tool path that is calculated by the simulation experiment causes the position deviation of tool tip $E$ to reduce from $4 \times 10^{-3}$ mm to $8 \times 10^{-6}$ mm and tool axis vector orientation $\varepsilon$ deviation to reduce from $3.5 \times 10^{-4}$ arc sec to $2.2 \times 10^{-4}$ arc sec. The compensated results demonstrate the effectiveness of the proposed method.

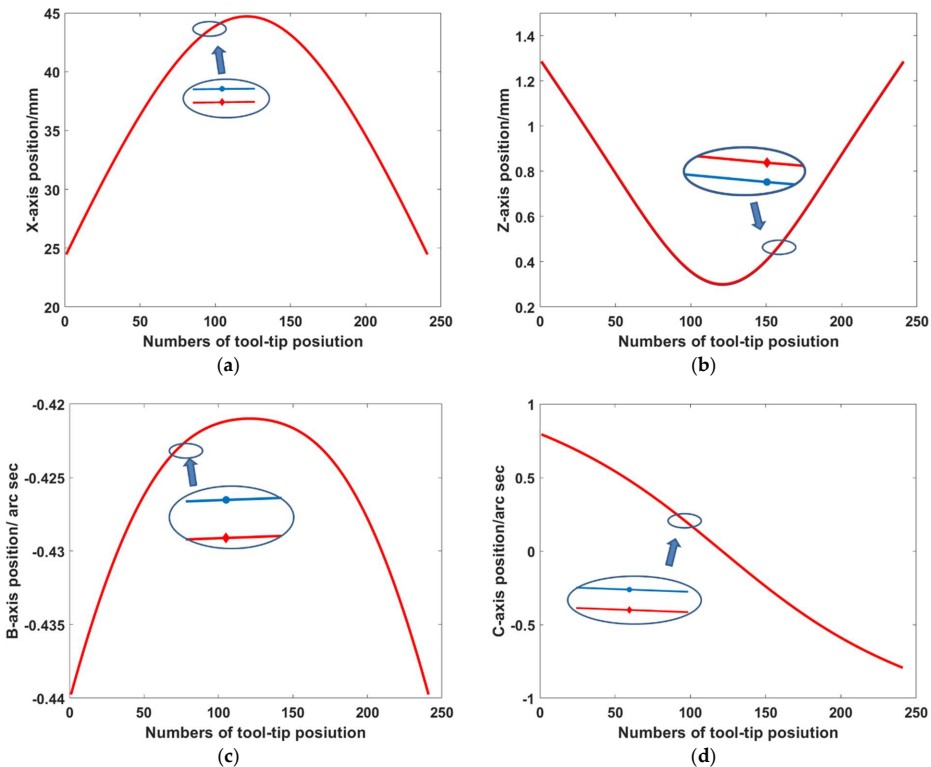

**Figure 13.** Axis position obtained from the kinematic structure with geometric error (the red line represents the original position of each axis calculated from the theoretical kinematic relation and the blue line represents the compensated trajectory position of each axis). (**a**) *X*-axis position considering geometric error. (**b**) *Z*-axis position considering geometric error. (**c**) *B*-axis position considering geometric error. (**d**) *C*-axis position considering geometric error.

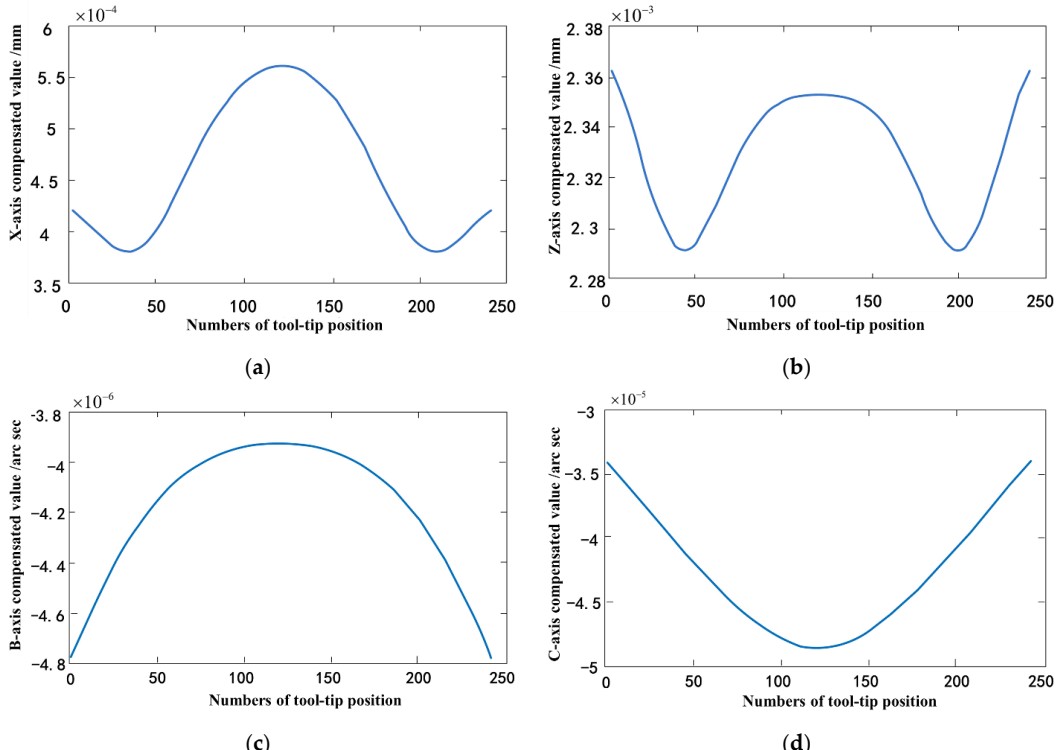

**Figure 14.** Axis compensated value from geometric error. (**a**) *X*-axis compensated value. (**b**) *Z*-axis compensated value. (**c**) *B*-axis compensated value. (**d**) *C*-axis compensated value.

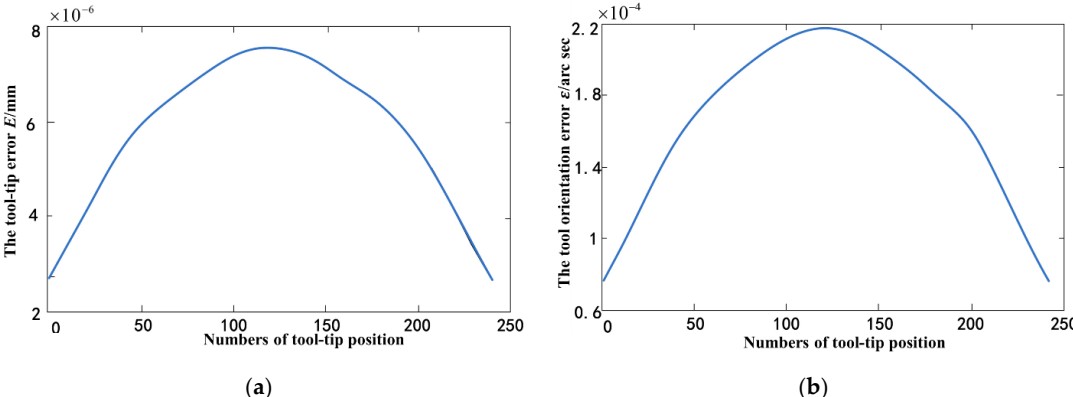

**Figure 15.** Deviation errors of tool-tip position and tool orientation after compensation. (**a**) Tool-tip position error after compensation. (**b**) Tool axis orientation error after compensation.

### 7. Conclusions

In this paper, the measured 6-Dof geometric errors that relate to the position of the machine tool were transformed into a homogeneous coordinate matrix. The geometric model was integrated into the kinematic equation of the machine tool as a means of studying the influence on the tool tip and tool-orientation trajectory. As solving the kinematic model of machine tool based on geometric error is quite difficult, the Newton iterative method was used. To solve it smoothly, the theoretical coordinates of each axis that were obtained from the inverse solution of the theoretical kinematic model were used as the initial value. In addition, the Jacobian matrix of the theoretical kinematic model of the machine tool replaced the Jacobian matrix that is based on the geometric error, thereby reducing calculation difficulty. Finally, using the four-axis machine tools developed in the laboratory and the processed three mirror lens, the influence that the geometric error has on the target trajectory and the error compensation calculation method based on Newton iteration were studied. The compensated tool path that was calculated from the simulation experiment caused the tool-tip position to deviate from $4 \times 10^{-3}$ mm to $8 \times 10^{-6}$ mm and the tool axis vector to deviate from $3.5 \times 10^{-4}$ arc sec to $2.2 \times 10^{-4}$ arc sec. The compensated tool path significantly reduced the relative position error.

**Author Contributions:** Writing—original draft preparation, G.Z.; writing—review and editing, S.J.; software, K.D.; software, Q.X.; data curation, Q.X.; writing—review and editing, and validation, Z.Z.; methodology, Z.Z.; L.L. All authors have read and agreed to the published version of the manuscript.

**Funding:** This research was supported by the National Natural Science Foundation of China (grant no. 51705120).

**Conflicts of Interest:** The authors declare that no conflict of interest exists in this paper.

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
