# Peer review of "Influence Analysis of Geometric Error and Compensation Method for Four-Axis Machining Tools with Two Rotary Axes"

_machines, doi:10.3390/machines10070586_

Round 1

Reviewer 1 Report

The four axis machine is modelled for geometric errors using the well known homogeneous transformation matcrices technique and the equations are developed for compensation model using newton iteration method. The Jacobian is formed which comes out as a non-square matric and the pseudo inversion technique is applied for taking the generalized inverse. To the understanding of the reviewer, the pseudo inverse provides some of the properties of matrix inverse (for the regular square matrix), which has a physical interpretation in the compensation model. It may work for half of the machine tool work space and may come up with different properties to the other half of the workspace of the machine tool. Therefore, it cannot be confidently said that psedoinversion technique for non-square matrix can be generally used for precision solutions without looking into its constituent properties as compared to regular matrix inverse. Therefore, the technique has its inherent underlying fundamental issues before it can be put directly to the application.  

Author Response

Responses to the Reviewers’ Comments and List of Amendments

Thank you for your time and comments concerning our manuscript entitled “Influence analysis of geometric error and compensation method for four-axis machining tools with two rotary axis (machines-1800460)”. These comments are all valuable and very helpful for improving our paper. We have revised the manuscript corresponding to the comments. Revised portions are highlighted in blue color in the manuscript and summarized as follows.

Comments from the reviewers:

Reviewer #1

Q1. The four axis machine is modelled for geometric errors using the well known homogeneous transformation matrices technique and the equations are developed for compensation model using newton iteration method. The Jacobian is formed which comes out as a non-square matric and the pseudo inversion technique is applied for taking the generalized inverse. To the understanding of the reviewer, the pseudo inverse provides some of the properties of matrix inverse (for the regular square matrix), which has a physical interpretation in the compensation model. It may work for half of the machine tool work space and may come up with different properties to the other half of the workspace of the machine tool. Therefore, it cannot be confidently said that psedo-inversion technique for non-square matrix can be generally used for precision solutions without looking into its constituent properties as compared to regular matrix inverse. Therefore, the technique has its inherent underlying fundamental issues before it can be put directly to the application.

Response: Thanks for the reviewer’s question. In solving linear equations group Ax=b, if the inverse matrices of A exits, the solution is x0= A-1b. When the number m of row of matrices A is greater than the number n column of matrices A, the linear Equations group Ax=b has no solution. However, a point x0 could be found to ensure the  minimum. According to the theory of Moore-penrosc psedo-inverse matrix, the point x0 could be calculated by x0=A+b, where A+ is Moore-penrosc psedo-inverse matrix of A.

When using Newton iteration to solve nonlinear equations groups , the inverse matric of  should be obtained. And, if the number m of equations f is greater than number n of unknowns , i.e. m>n, the inverse matric of  does not exist and the  has no solution. As the essence of Newton iteration is local linearization, the point  to ensure the  minimum could be calculated literately by Moore-penrosc psedo-inverse matrix  of  based on Newton iteration method. To be noted that, the solution found by this method is the near solution as the  has no solution when m>n.

In order to make this statement more clearly, we have revised the manuscript. The updated texts are copied as follows and also highlighted in blue color in the revised manuscript.

On the line 328 of 9: “(To be noted that, the solution found by this method is the near solution as the  has no solution when m>n.)”

Reviewer 2 Report

The paper deals with the Influence analysis of geometric error and compensation
method for four-axis machining tools with two rotary axis.
According to the reviewer, the paper is worth publishing at Machines Journal,
but some corrections are needed and then the paper can be accepted for publication in the journal.
While the authors have made considerable research effort,
the presentation of the paper and the results must be proved.
Additionally make the following corrections to the manuscript:

Comment 1
Line 7
Robotics &Collaborative
The authors should replace (insert a space)
Robotics & Collaborative

Comment 2
Line 58
Peng Niu et al. [14]
The authors should replace
Niu et al. [14]

Line 59
global sensitivity analysis (GSA) method
The authors should replace
Global Sensitivity Analysis (GSA) method

Line 62
Hareendran Manikandan et al. [15]
The authors should replace
Manikandan et al. [15]

Line 65
Jinwei Fan et al.
The authors should replace
Fan et al.

Line 68
Chuandong Li et al. [17]
The authors should replace
Li et al. [17]

Line 70
device reached 91.8%
The authors must format the paper according to the journal's instructions
The text is out of limits.

Line 71
Xuemin Zhong et al. [18]
The authors should replace
Zhong et al. [18]

Line 81
Ruijun Liang et al. [20]
The authors should replace
Liang et al. [20]

Line 84
Jun Zha et al. [21]
The authors should replace
Zha et al. [21]

Lines 86 - 87
Kodai Nagayama [22]
The authors should replace
Nagayama et al. [22]

Line 89
Zhenjiu Zhang et al. [23]
The authors should replace
Zhang et al. [23]

Line 91
Toru Fujimori et al. [24]
The authors should replace
Toru Fujimori et al. [24]

Line 93
Ying-Chen Lu et al.
The authors should replace
Lu et al.

Line 98
Zhonghui Luo et al. [26]
The authors should replace
Luo et al. [26]

Comment 3
Figure 1b
The description of the Figure 1b is considered incomplete.
The authors must gine more details.

Comment 4
Lines 161 - 165
Lines 252 - 258
The authors must format the tables so that the numbers in the columns must be aligned.

Comment 5
Lines 187 - 188
The error of any kinematic pair on
the kinematic chain is assumed to not affect other kinematic pairs.
The word "kinematic" is used 3 times.
The authors must rephrase.

Lines 187 - 188
The machine tool controller sends the
motion position command to the moving axis and the actual position of the moving part
of the moving axis is measured by the mirror group that is installed on the moving part
of the moving axis.
The word "moving" is used 4 times.
The authors must rephrase.

Comment 6
The authors must increase the visibility of the Figure 3.

Comment 7
Line 287
Eq.(13)
The authors should replace (insert a space)
Eq. (13)

Line 294
in Eq.(9),
The authors should replace (insert a space)
in Eq. (9),

Comment 8
Line 318
The authors must give more details for the equipment (type, model,...).

Comment 9
The Figure must be accompanied on the same page as the Figure's title (Figures 6, 8, 9 and 13).
The authors must format the paper.

Comment 10
Line 346
The authors must give more details for the equipment (type, model,...).

Comment 11
Lines 355 - 356
Line 366
The authors must format the paper.

Comment 12
Figure 12
The authors must explain why the red line is above from the blue line only in the case of the Z-axis position considering geometric error.

Comment 13
References
According to thw instructions
https://www.mdpi.com/journal/machines/instructions
References should be described as follows, depending on the type of work:

Journal Articles:
1. Author 1, A.B.; Author 2, C.D. Title of the article. Abbreviated Journal Name Year, Volume, page range.

Increase the number of the reference papers including (primarily) from MDPI Journals
The authors use 0 paper from Machines Journal / 2 MDPI Journals / 27 papers from journals (References)
Τhe number for papers from MDPI journals is considered insufficient (in reviewer's opinion).

Author Response

Responses to the Reviewers’ Comments and List of Amendments (Reviewer 2)

Thank you for your time and comments concerning our manuscript entitled “Influence analysis of geometric error and compensation method for four-axis machining tools with two rotary axis (machines-1800460)”. These comments are all valuable and very helpful for improving our paper. We have revised the manuscript corresponding to the comments. Revised portions are highlighted in blue color in the manuscript and summarized as follows.

Comments from the reviewers:

Reviewer #2

Q1. Comment 1 Line 7
Robotics &Collaborative
The authors should replace (insert a space)
Robotics & Collaborative

Response: Thanks for the reviewer’s suggestion. We have inserted a space in the sentence.

Q2. Comment 2 Line 58

Peng Niu et al. [14]

The authors should replace

Niu et al. [14]

Response: Thanks for the reviewer’s suggestion. We have replaced the name of the author.

Q3. Line 59
global sensitivity analysis (GSA) method
The authors should replace
Global Sensitivity Analysis (GSA) method

Response: Thanks for the reviewer’s suggestion. We have revised this part.

Q4. Line 62
Hareendran Manikandan et al. [15]
The authors should replace
Manikandan et al. [15]

Response: Thanks for the reviewer’s suggestion. We have replaced the name of the author.

Q5. Line 65
Jinwei Fan et al.
The authors should replace
Fan et al.

Response: Thanks for the reviewer’s suggestion. We have replaced the name of the author.

Q6. Line 68
Chuandong Li et al. [17]
The authors should replace
Li et al. [17]

Response: Thanks for the reviewer’s suggestion. We have replaced the name of the author.

Q7. Line 70
device reached 91.8%
The authors must format the paper according to the journal's instructions
The text is out of limits.

Response: Thanks for the reviewer’s suggestion. We have revised the manuscript.

Q8. Line 71
Xuemin Zhong et al. [18]
The authors should replace
Zhong et al. [18]

Response: Thanks for the reviewer’s suggestion. We have replaced the name of the author.

Q9. Line 81
Ruijun Liang et al. [20]
The authors should replace
Liang et al. [20]

Response: Thanks for the reviewer’s suggestion. We have replaced the name of the author.

Q10. Line 84
Jun Zha et al. [21]
The authors should replace
Zha et al. [21]

Response: Thanks for the reviewer’s suggestion. We have replaced the name of the author.

Q11. Lines 86 - 87
Kodai Nagayama [22]
The authors should replace
Nagayama et al. [22]

Response: Thanks for the reviewer’s suggestion. We have replaced the name of the author.

Q12. Line 89
Zhenjiu Zhang et al. [23]
The authors should replace
Zhang et al. [23]

Response: Thanks for the reviewer’s suggestion. We have replaced the name of the author.

Q13. Line 91
Toru Fujimori et al. [24]
The authors should replace
Toru Fujimori et al. [24]

Response: Thanks for the reviewer’s suggestion. We have replaced the name of the author.

Q14. Line 93
Ying-Chen Lu et al.
The authors should replace
Lu et al.

Response: Thanks for the reviewer’s suggestion. We have replaced the name of the author.

Q15. Line 98
Zhonghui Luo et al. [26]
The authors should replace
Luo et al. [26]

Response: Thanks for the reviewer’s suggestion. We have replaced the name of the author.

Q15. Figure 1b

The description of the Figure 1b is considered incomplete.

The authors must give more details.

Response: Thanks for the reviewer’s suggestion. We have revised the description of the Figure 1b. The updated texts are copied as follows and also highlighted in blue color in the revised manuscript.

Q16. Lines 161 - 165
Lines 252 - 258
The authors must format the tables so that the numbers in the columns must be aligned.

Response: Thanks for the reviewer’s suggestion. Lines 161 – 165 and Lines 252 – 258 are the descriptions of Eqs. (2) and (8). In order to clearly explain the elements of the matrix, the rows and columns of matrix are also expressed. They are not tables. However, in order to make the matrix arrangement more orderly, the rows and columns have been aligned.

Q17. Lines 187 - 188

The error of any kinematic pair on the kinematic chain is assumed to not affect other kinematic pairs.

The word "kinematic" is used 3 times.

The authors must rephrase.

Response: Thanks for the reviewer’s suggestion. We have rephrased the sentence. The updated texts are copied as follows and also highlighted in blue color in the revised manuscript.

On the line 185 of 5: “The error of any kinematic pair is assumed to not affect other pairs.”

Q18. Lines 187 - 188

The machine tool controller sends the motion position command to the moving axis and the actual position of the moving part of the moving axis is measured by the mirror group that is installed on the moving part of the moving axis.

The word "moving" is used 4 times.

The authors must rephrase.

Response: Thanks for the reviewer’s suggestion. We have rephrased the sentence. The updated texts are copied as follows and also highlighted in blue color in the revised manuscript.

On the line 209 of 5: “The machine tool controller sends the position command to the moving axis and the actual position of the moving part of the axis is measured by the mirror group that is installed on the moving part of the axis.”

Q19. The authors must increase the visibility of the Figure 3.

Response: Thanks for the reviewer’s suggestion. We have revised the figure.

Q20. Line 287

Eq.(13)

The authors should replace (insert a space)

Eq. (13)

Line 294

in Eq.(9),

The authors should replace (insert a space)

in Eq. (9),

Response: Thanks for the reviewer’s suggestion. We have revised the equations.

Q20. Line 318

The authors must give more details for the equipment (type, model,...).

Response: Thanks for the reviewer’s suggestion. We have provided the type and model of the four-axis machine tools in the revised manuscript. The updated texts are copied as follows and also highlighted in blue color in the revised manuscript.

On the line 360 of 9: “The machine tool combines with two linear axis and two rotatory axes. The two linear axes are ABL8000 Series linear motor stage with air bearing developed by Aerotech. The B-axis is ABRS Series air-bearing rotary stage also developed by Aerotech company. The C-axis is ultraprecision work-holding spindles (SP150 High performance) developed by Precitech company.”

Q21. The Figure must be accompanied on the same page as the Figure's title (Figures 6, 8, 9 and 13).
The authors must format the paper.

Response: Thanks for the reviewer’s suggestion. We have revised the manuscript to ensure the Figure's title accompany with the figure in the same page.

Q22. Line 346

The authors must give more details for the equipment (type, model,...).

Response: Thanks for the reviewer’s suggestion. We have provided the type and model of the laser interferometer in the revised manuscript. The updated texts are copied as follows and also highlighted in blue color in the revised manuscript.

On the line 399 of 11: “(The laser interferometer is XL-80 laser interferometer system of the Renishaw company which has a lens group for measuring linear displacement, speed, angle (pitch and torsion), straightness, flatness, perpendicularity and parallelism. [27])”

Q23. Lines 355 - 356
Line 366
The authors must format the paper.

Response: Thanks for the reviewer’s suggestion. We have revised the manuscript.

Q24. Figure 12

The authors must explain why the red line is above from the blue line only in the case of the Z-axis position considering geometric error.

Response: Thanks for the reviewer’s suggestion. Since the geometric error is small relative to the movement of the axis, we use a local enlarged view to illustrate. The red line is above from the blue line only in the case of the Z-axis position comes from the results and the measured geometric errors. This is normal and in other position of different axes, it is also possible that the red line is above the blue line.

Q25. Comment 13

References

According to the instructions

https://www.mdpi.com/journal/machines/instructions

References should be described as follows, depending on the type of work:

Journal Articles:

  1. Author 1, A.B.; Author 2, C.D. Title of the article. Abbreviated Journal Name Year, Volume, page range.

Increase the number of the reference papers including (primarily) from MDPI Journals

The authors use 0 paper from Machines Journal / 2 MDPI Journals / 27 papers from journals (References)

The number for papers from MDPI journals is considered insufficient (in reviewer's opinion).

Response: Thanks for the reviewer’s suggestion. We have revised the manuscript and some papers from the Machines Journal and MDPI Journals are added. The added papers are provided as follow and also highlighted in blue color in the revised manuscript.

On the line 621 of 19: “23.     Song, Z.; Ding, S.; Chen, Z.; Lu, Z.; Wang, Z. High-Efficient Calculation Method for Sensitive PDGEs of Five-Axis Reconfigurable Machine Tool. Machines 2021, 9, 84. https://doi.org/10.3390/machines9050084”

On the line 626 of 19: “25.     Ding, W.; Song, Z.; Ding, S. Investigation on Structural Mapping Laws of Sensitive Geometric Errors Oriented to Remanufacturing of Three-Axis Milling Machine Tools. Machines 2022, 10, 341. https://doi.org/10.3390/machines10050341”

On the line 638 of 19: “29.     Lu, H.; Cheng, Q.; Zhang, X.; Liu, Q.; Qiao, Y.; Zhang, Y. A Novel Geometric Error Compensation Method for Gantry-Moving CNC Machine Regarding Dominant Errors. Processes 2020, 8, 906. https://doi.org/10.3390/pr8080906”

Reviewer 3 Report

ü  Axes and axis, the difference between spelling, correct it, make it uniform.

ü  Abstract needs revision, The abstract should have one sentence per each: context and background, motivation, hypothesis, methods, results, and conclusions.

ü  Avoid lumping of ref., e.g., [3-5], [6-8], lines 29-31, and write the contribution of each.

ü  Line 48; D-H method, write full form.

ü  Line 46 to 48, different methods names can be supported with reference/citation to individual methods; please take care.

ü  Explain the RLLLR five-axis machine tool,

ü  The problem formulation statement is missing. Write down the organization of the rest of the paper after the problem formulation statement and the study’s objective.

ü  Define clearly research questions and objectives.

ü  Why was the Newton iteration method chosen? And Compensate geometric error. The importance of the method is missing while utilizing review studies.

ü  The link between Sections is missing. It can be compensated by drawing a workflow diagram. Section 2 Materials and Methods need to be added to define the procedure, measuring, and other methods used.

ü  Sources of Equations needed, add references. Applied methods are without citations.

ü  Add data Labels in Figure 7, and a photographic view must present what is linked with the study.

ü  What standards were used for measurement?

ü  Line 348; Due to laboratory conditions, the C-axis is not measured. The meaning is not clear.

o   Conclusions:

ü  In this paper, the measured 6-Dof geometric errors that relate to the position of the 471 machine tool are transformed into a homogeneous coordinate matrix.

o   I don’t find the 6-Dof in the whole paper, and the first line of the conclusion starts. I think authors are not clear about what they want to covey.

o   Presentation is very poor, need all rearrangement of Sections.

ü  Why is the topic important (or why do you study it)? What are the research questions? What has been studied? What are your contributions?

ü  Why is it to propose this particular method? The major defect of this study is the debate or argument is not clearly stated in the introduction session.

ü  Please explain the problem you want to solve and the study’s contributions in the abstract.

Author Response

Responses to the Reviewers’ Comments and List of Amendments (Reviewer 3)

Thank you for your time and comments concerning our manuscript entitled “Influence analysis of geometric error and compensation method for four-axis machining tools with two rotary axis (machines-1800460)”. These comments are all valuable and very helpful for improving our paper. We have revised the manuscript corresponding to the comments. Revised portions are highlighted in blue color in the manuscript and summarized as follows.

Comments from the reviewers:

Reviewer #3

Q1. Axes and axis, the difference between spelling, correct it, make it uniform.

Response: Thanks for the reviewer’s suggestion. We have revised the manuscript to make it uniform.

Q2. Abstract needs revision, the abstract should have one sentence per each: context and background, motivation, hypothesis, methods, results, and conclusions.

Response: Thanks for the reviewer’s suggestion. The motivation and results has been added in Abstract. The abstract has been revised according to reviewer’s suggestion. The updated texts are copied as follows and also highlighted in blue color in the revised manuscript.

On the line 12 of 1: “Four-axis machine tools with two rotary axes are widely used in the machining of complex parts. However, due to an irregular kinematic relationship and non-linear kinematic function with geometric error, it is difficult to analyze the influence the geometry error of each axis has and to compensate for such a geometry error. In this article, an influence analysis method of geometric error based on the homogeneous coordinate transformation matrix and a compensation method is developed using the Newton iterative method. Geometric errors are characterized by a homogeneous coordinate transformation matrix in the proposed method, and an error matrix is integrated into the kinematic model of the four-axis machine tool as a means of studying the influence the geometric error of each axis has on the tool path. Based on the kinematic model of the four-axis machine tool considering the geometric error, a comprehensive geometric error compensation calculation model based on the Newton iteration is then constructed for calculating the tool path as a means of compensating for the geometric error. Ultimately, the four-axis machine tool with a curve tool path for an off-axis optical lens is chosen for verification of the proposed method. The results shown that the proposed method can significantly improve the machining accuracy.”

Q3. Avoid lumping of ref., e.g., [3-5], [6-8], lines 29-31, and write the contribution of each.

Response: Thanks for the reviewer’s suggestion. We have revised the manuscript. The updated texts are copied as follows and also highlighted in blue color in the revised manuscript.

On the line 34 of 1: “Wu et al. [3] proposed a robust design method for optimizing the static accuracy of a vertical machining center to make the machining accuracy meet design requirements. Niu et al. [4] provided a new analysis method for evaluating machining accuracy reliability based on the nonlinear correlation between errors. Li et al. [5] overviewed the thermal error modeling methods that had been researched and applied in the past ten years.”

On the line 41 of 1: “To improve the accuracy of error recognition, Wei et al. [6] provided an overview of the current research algorithms. Lin et al. [7] provided a geometric error modeling method for five-axis CNC machine tools based on differential transformation method. Geng et al. [8] summarized state-of-the-art research in the calibration of geometric errors of ultra-precision machine tools (UPMTs). Compared to the mature method for traditional precision machine tools, the increasing use of UPMTs has shown different characteristics in error modelling, measurement, and compensation.”

Q4. Line 48; D-H method, write full form.

Response: Thanks for the reviewer’s suggestion. We have written full form of D-H. The updated texts are copied as follows and also highlighted in blue color in the revised manuscript.

On the line 62 of 2: “Geometric error modeling methods have been developed from a variety of perspectives and include the triangular geometry method, vector method, rigid body kinematics, Denavit-Hartenberg (D-H) method, error matrix method, quadratic analysis method, and multi-body system kinematic model.”

Q5. Line 46 to 48, different methods names can be supported with reference/citation to individual methods; please take care.

Response: Thanks for the reviewer’s suggestion. The reference related with these methods are provided. The updated texts are copied as follows and also highlighted in blue color in the revised manuscript.

On the line 61 of 2: “Geometric error modeling methods have been developed from a variety of perspectives and include the triangular geometry method [11], rigid body kinematics [12], De-navit-Hartenberg (D-H) method [13], error matrix method [14], and quadratic analysis method [15].”

Q6. Explain the RLLLR five-axis machine tool,

Response: Thanks for the reviewer’s suggestion. the explain for the are provide in the revised manuscript. The updated texts are copied as follows and also highlighted in blue color in the revised manuscript.

On the line 102 of 3: “The RLLLR five-axis machine tool is constructed by two rotatory axes and three linear axes. The kinematic chain from workpiece coordinate system to the tool coordinate system is the rotatory axis, the linear axis, the linear axis, the linear axis and the rotatory axis successively.”

Q7. The problem formulation statement is missing. Write down the organization of the rest of the paper after the problem formulation statement and the study’s objective. Define clearly research questions and objectives.

Response: Thanks for the reviewer’s suggestion. The problem formulation statement is provided and the organization of the rest of the paper is also presented in the instruction.

The research question is that for four-axis machine tools with two rotatory axes, it is difficult to analyze the influence the geometry error of each axis has and to compensate the geometry error considering its irregular kinematic relationship compared with the traditional five axis machine tools. And the kinematic model of machine tools with geometric error is a complex highly nonlinear equation, the Jacobian matrix is non-spare matrix due to the irregular kinematic relationship which increase the difficulty of the analysis and compensation process.

The object is to find a proper analysis and compensation process by design the calculating process of Newton iteration method and to significantly improve the machining accuracy.

The updated texts are copied as follows and also highlighted in blue color in the revised manuscript.

On the line 126 of 3: “The independently developed four-axis machine tool is taken as the research object for this paper. In the traditional five axis machine tool compensating process, the position error of tool tip is compensated by the linear axes of the machine tool and the orientation error of the cutting tool is compensated by the rotating axes of the machine tool. However, for four-axis machine tools with two rotatory axes, it is difficult to analyze the influence the geometry error of each axis has and to compensate the geometry error considering its irregular kinematic relationship compared with the traditional five axis machine tools.

Due to the kinematic model of machine tools with geometric error is a highly nonlinear complex equation, the Newton iterative method is used for solving nonlinear equations. As the irregular kinematic relationship, the Jacobian matrix is non-spare matrix which increase the difficulty of the iteration process. To solve them smoothly, the pseudo-inverse matrix of the Jacobian matrix is used to find the near-solution. At the same time, the theoretical coordinates of each axis that are obtained from the inverse solution of the theoretical kinematic model are taken as the initial value, and the Jacobian matrix of the theoretical kinematic model of the machine tool replaces the Jacobian matrix based on geometric error, thereby reducing calculation difficulty. The four-axis machining platform with two linear motors and direct drive turntables is used for verifying the comprehensive error analysis and compensation method, significantly im-proving machining accuracy. The rest of this paper is organized as follows: In Section 2, Theoretical kinematic relationship for four-axis machine tools is provided. The geo-metric error is modeled in Section 3. In Section 4, the actual kinematic relationship is established with the geometric errors. The geometric error is compensated based on the Newton iteration in Section 5. Experiment and simulation of error compensation are presented in Section 6, and finally the paper is concluded in Section 7.”

Q8. Why was the Newton iteration method chosen? And Compensate geometric error. The importance of the method is missing while utilizing review studies.

Response: Thanks for the reviewer’s suggestion. Due to the kinematic model of machine tools with geometric error is a highly nonlinear complex equation, the Newton iterative method is used for solving nonlinear equations. At the same, other gradient based algorithms for solving nonlinear equations can also be used and some special treatments (such as using pseudo-inverse matrix of the Jacobian matrix in the iterative process and using the theoretical coordinates obtained from the inverse solution of the theoretical kinematic model as the initial value.) for the calculating process is also same as the proposed method.

The updated texts are copied as follows and also highlighted in blue color in the revised manuscript.

On the line 133 of 3: “Due to the kinematic model of machine tools with geometric error is a complex highly nonlinear equation, the Newton iterative method is used for solving nonlinear equations. As the irregular kinematic relationship, the Jacobian matrix is non-spare matrix which increase the difficulty of the iteration process. To solve them smoothly, the pseudo-inverse matrix of the Jacobian matrix is used to find the near-solution.”

Q9. The link between Sections is missing. It can be compensated by drawing a workflow diagram. Section 2 Materials and Methods need to be added to define the procedure, measuring, and other methods used.

Response: Thanks for the reviewer’s suggestion. A workflow diagram was provided to reflect the link between Sections. The flowchart is provided as follow and also presented in the revised manuscript.

Figure 1. The flowchart of the proposed method.

The innovation of the proposed method is mainly in applied mathematical methods, no special materials are needed in the proposed method. And the type and model of the four-axis machine tools and laser interferometer are also provided. In order to make the procedure more clearly, the produced is added as reviewer suggestion in the end of the instruction.

The updated texts are copied as follows and also highlighted in blue color in the revised manuscript.

On the line 150 of 3: “The flowchart is also provided in Figure 1. From the flowchart, the laser interferometer is used firstly for detecting and analyzing the position of geometric errors of the axes of the four-axis machining platform through many experiments. Secondly, the multi-body system theory is used for integrating the error matrix into the kinematic model of the machine tool as a means of studying the influence the geometric error of each axis has on the tool path. The nonlinear coupling characteristics of the transformation matrix from tool coordinate system to workpiece coordinate system can then be analyzed. Finally, a comprehensive error kinematic model of the our-axis machining platform considering geometric error is established.”

Q10. Sources of Equations needed, add references. Applied methods are without citations.

Response: Thanks for the reviewer’s suggestion. In the proposed method, most of the equations are modeling the kinematic relationship. The Newton iteration calculating process is cited from “An Introduction to Numerical Analysis”. The reference is added in revised manuscript.

Q11. Add data Labels in Figure 7, and a photographic view must present what is linked with the study.

Response: Thanks for the reviewer’s suggestion. the axis moving direction labels are added in figure 7 as reviewer suggestion.

The updated figures are copied as follows and also highlighted in blue color in the revised manuscript.

On the line 407 of 11:

(a)                             (b)                            (c)

Figure 7. Measuring process. (a) Measuring process for the Z-axis; (b) Measuring process for the X-axis; (c) Measuring process for the B-axis.

Q12. What standards were used for measurement?

Response: Thanks for the reviewer’s suggestion. the measurement is mainly based on Standard: ISO 230. Especially based on ISO 230-1:20212, it is that geometric accuracy of machines operating under no-load or quasi-static conditions.

The updated texts are copied as follows and also highlighted in blue color in the revised manuscript.

On the line 399 of 11: “As Figure 8 shows, the laser interferometer (The laser interferometer is XL-80 laser interferometer system of the Renishaw company which has a lens group for measuring linear displacement, speed, angle (pitch and torsion), straightness, flatness, perpendicularity and parallelism. [35]) is used for repeatedly measuring positioning error, straightness error, yaw error, and pitch error for the X-axis, Z-axis, and B-axis based on Standard: ISO 230.”

Q13. Due to laboratory conditions, the C-axis is not measured. The meaning is not clear.

Response: Thanks for the reviewer’s question. Just as this sentence statement, the geometric errors of C-axis are not measured. However, the geometric errors from other axes are compensated by C-axis.

In order to make this sentence more clearly, this sentence is revised. The updated texts are copied as follows and also highlighted in blue color in the revised manuscript.

On the line 404 of 11: “Due to laboratory conditions, the geometric errors of C-axis are not measured. How-ever, the geometric errors from other axes are compensated by C-axis.”

Q14. In this paper, the measured 6-Dof geometric errors that relate to the position of the machine tool are transformed into a homogeneous coordinate matrix. I don’t find the 6-Dof in the whole paper, and the first line of the conclusion starts. I think authors are not clear about what they want to covey. Presentation is very poor, need all rearrangement of Sections.

Response: The first statement of the 6-Dof geometric errors is presented in Line 228 of Section 3 “There are six elements to the motion error of any motion pair and the movement and rotation relative to the X, Y, and Z axes, which are the geometric errors of machine tools, mainly include positioning error, straightness error, rolling angle error, and yaw angle error.” And how to transfer the measured 6-Dof geometric errors to the homogeneous coordinate matrix is also provided in Eq. (6).

We are sorry for the poor presentation of this part and we also revised the paper as all reviewers’ kindly suggestion. We also try our best to make this paper clearer.

Q15. Why is the topic important (or why do you study it)? What are the research questions? What has been studied? What are your contributions?

Response: Thanks for the reviewer’s question. Geometric error plays an important role in the machining accuracy of machine tools. And many articles in recent years as presented in Introduction provided many different modelling and compensating method. Therefore, it is important to study how to model and compensate the geometric error for four-axis machine tool.

In the traditional five axis machine tool compensating process, the position error of tool tip is compensated by the linear axes of the machine tool and the orientation error of the cutting tool is compensated by the rotating axes of the machine tool. However, for four-axis machine tools with two rotatory axes, it is difficult to analyze the influence the geometry error of each axis has and to compensate the geometry error considering its irregular kinematic relationship compared with the traditional five axis machine tools.

Due to the kinematic model of machine tools with geometric error is a complex highly nonlinear equation, the Newton iterative method is provided for compensating the geometric error in this paper. Considering the irregular kinematic relationship, the Jacobian matrix for the four-axis machine tools is non-spare matrix which increase the difficulty of the iteration process. To solve them smoothly, the pseudo-inverse matrix of the Jacobian matrix is used to find the near-solution. At the same time, the theoretical coordinates of each axis that are obtained from the inverse solution of the theoretical kinematic model are taken as the initial value, and the Jacobian matrix of the theoretical kinematic model of the machine tool replaces the Jacobian matrix based on geometric error, thereby reducing calculation difficulty.

Q16. Why is it to propose this particular method? The major defect of this study is the debate or argument is not clearly stated in the introduction session.

Response: Thanks for the reviewer’s question. In the traditional five axis machine tool compensating process, the position error of tool tip is compensated by the linear axes of the machine tool and the orientation error of the cutting tool is compensated by the rotating axes of the machine tool. However, for four-axis machine tools with two rotatory axes, it is difficult to analyze the influence the geometry error of each axis has and to compensate the geometry error considering its irregular kinematic relationship compared with the traditional five axis machine tools.

The introduction session has been revised as reviewer suggestion. The updated texts are copied as follows and also highlighted in blue color in the revised manuscript.

On the line 127 of 3: “In the traditional five axis machine tool compensating process, the position error of tool tip is compensated by the linear axes of the machine tool and the orientation error of the cutting tool is compensated by the rotating axes of the machine tool. However, for four-axis machine tools with two rotatory axes, it is difficult to analyze the influence the geometry error of each axis has and to compensate the geometry error considering its irregular kinematic relationship compared with the traditional five axis machine tools.”

Q17. Please explain the problem you want to solve and the study’s contributions in the abstract.

Response: Thanks for the reviewer’s question. Due to an irregular kinematic relationship and non-linear kinematic function with geometric error, it is difficult to analyze the influence the geometry error of each axis has and to compensate for such a geometry error. In this article, an influence analysis method of geometric error based on the homogeneous coordinate transformation matrix and a compensation method is developed using the Newton iterative method. Ultimately, the four-axis machine tool with a curve tool path for an off-axis optical lens is chosen for verification of the proposed method. The results shown that the proposed method can significantly improve the machining accuracy. The abstract has been revised according to reviewer’s suggestion. The updated texts are copied as follows and also highlighted in blue color in the revised manuscript.

On the line 12 of 1: “Four-axis machine tools with two rotary axes are widely used in the machining of complex parts. However, due to an irregular kinematic relationship and non-linear kinematic function with geometric error, it is difficult to analyze the influence the geometry error of each axis has and to compensate for such a geometry error. In this article, an influence analysis method of geometric error based on the homogeneous coordinate transformation matrix and a compensation method is developed using the Newton iterative method. Geometric errors are characterized by a homogeneous coordinate transformation matrix in the proposed method, and an error matrix is integrated into the kinematic model of the four-axis machine tool as a means of studying the influence the geometric error of each axis has on the tool path. Based on the kinematic model of the four-axis machine tool considering the geometric error, a comprehensive geometric error compensation calculation model based on the Newton iteration is then constructed for calculating the tool path as a means of compensating for the geometric error. Ultimately, the four-axis machine tool with a curve tool path for an off-axis optical lens is chosen for verification of the proposed method. The results shown that the proposed method can significantly improve the machining accuracy.”

Round 2

Reviewer 1 Report

The paper is improved and explained upto the satisfactory level. 

Reviewer 3 Report

Comments clearly stated and manuscript updated